# An oral, liver-restricted LXR inverse agonist for dyslipidemia: preclinical development and phase 1 trial

Xiaoxu Li [1], Giorgia Benegiamo [1], Archana Vijayakumar[2], Natalie Sroda[2], Masaki Kimura[3,4], Ryan S. Huss [2], Steve Weng[2], Eisuke Murakami[2], Brian J. Kirby[2], Giacomo V. G. von Alvensleben[1], Claus Kremoser[5], Edward J. Gane[6], Takanori Takebe [3,4,7], Robert P. Myers[2], G. Mani Subramanian [2] ✉ & Johan Auwerx [1] ✉

Despite advances in lipid-lowering treatment, atherosclerotic cardiovascular disease remains the leading cause of mortality, underscoring the need to address residual risk. Targeting both the synthesis and clearance of triglyceride (TG)-rich lipoproteins is a promising approach. Liver X receptor (LXR) repression can reduce plasma TG and cholesterol and improve insulin sensitivity by suppressing de novo lipogenesis and intestinal lipid absorption and enhancing clearance of TG-rich lipoproteins, but its clinical utility remains unexplored. Here we demonstrate the role of LXR inverse agonists in lipid metabolism and metabolic diseases in preclinical models and humans. Given concerns that systemic LXR repression may impair reverse cholesterol transport, we developed TLC-2716, an orally administered, gut- and liver-restricted LXR inverse agonist. In human liver organoids modeling steatohepatitis, TLC-2716 reduced lipid accumulation and suppressed inflammation and fibrotic gene expression. In a randomized, placebo-controlled phase 1 clinical trial, 14-day treatment with TLC-2716 was well tolerated (primary endpoints) and resulted in placebo-adjusted reductions up to 38.5% in plasma TG and 61% in postprandial remnant cholesterol (secondary endpoints). In conclusion, these results highlight the tolerability and therapeutic potential of TLC-2716 as a treatment for managing dyslipidemia and reducing residual atherosclerotic cardiovascular disease risk in humans. ClinicalTrials.gov identifier: NCT05483998.

Dyslipidemia encompasses a broad spectrum of lipid abnormalities, including hypertriglyceridemia (serum triglyceride (TG) > 150 mg dl⁻¹, a common dyslipidemia in adults[1]), elevated low-density lipoprotein cholesterol (LDL-C) and reduced high-density lipoprotein cholesterol (HDL-C). Excessive circulating TG contributes to metabolic disorders, including acute pancreatitis[2], atherosclerotic cardiovascular disease (ASCVD)[3] and metabolic dysfunction-associated steatotic liver disease (MASLD)[4]. Elevated remnant cholesterol (RC), which represents the cholesterol in TG-rich lipoproteins, is also an independent risk factor for ASCVD[5]. Currently, management of severe hypertriglyceridemia (TG ≥ 500 mg dl⁻¹) focuses on lifestyle modification and fibrates as first-line therapy[6]. Given the increasing prevalence of hypertriglyceridemia, research into more effective treatment strategies remains essential.

Hypertriglyceridemia results from an imbalance between the production and release of TG-rich lipoproteins from the liver (very

low-density lipoprotein (VLDL)) and intestine (chylomicrons) and lipolytic removal of TG from these lipoproteins and their remnants[5]. Lipoprotein lipase (LPL) is the main enzyme involved in hydrolysis of TG-rich lipoproteins, and its activity is inhibited by apolipoprotein C3 (ApoC3) and angiopoietin-like proteins (for example, ANGPTL3 and ANGPTL4); therapies that inhibit ANGPTLs and ApoC3 to reduce plasma TG are under investigation[7–9]. Furthermore, inhibiting de novo lipogenesis (DNL)[10] provides another strategy to reduce serum TG and potentially tissue inflammation attributable to lipotoxic mediators[11].

The liver X receptors (LXRs; LXRα (encoded by *NR1H3*) and LXRβ (encoded by *NR1H2*)) are members of the nuclear hormone receptor superfamily and key transcriptional regulators of systemic lipid metabolism[12,13]. LXR activation increases liver and plasma TG and LDL-C in animal models[14–17] and humans[18] by upregulating DNL genes[10], including *SREBF1*, *ACACA* and *FASN*, and suppressing hepatic LPL activity by inducing *ANGPTL3* and *APOC3* (ref. 7). By contrast, repression of LXR activity in the liver reduces hepatic DNL and, consequently, plasma and liver TG and improves hypertriglyceridemia and related comorbidities in preclinical models[19]. Therapies that inhibit LXR target genes involved in lipogenesis (for example, *ACACA* and *FASN*) and lipid clearance (for example, *ANGPTL3* and *APOC3*) show promising safety and efficacy in clinical trials[20]. Repression of LXR systemically carries potential risks, including the impairment of reverse cholesterol transport (RCT), the efflux of cholesterol from peripheral cells such as macrophages, which could promote atherogenesis and negatively impact cardiometabolic health[21]. In fact, LXR agonists increase *ABCA1* expression and RCT[22,23] and reduce atherosclerosis in mice[24], but their clinical development was hindered by parallel increases in plasma TG and LDL-C. Together, these data formed the basis for the development of LXR inverse agonists that have shown beneficial effects on hyperlipidemia[25], alcoholic liver disease[26,27] or MASLD[28] preclinically. However, to date, no LXR inverse agonist has been evaluated in humans.

Here, we leveraged human genetic data, dysmetabolic rodent models, humanized experimental models, toxicology studies and a phase 1 clinical trial to evaluate the tolerability and efficacy of TLC-2716, an orally administered LXR inverse agonist, which to date has not been tested in humans[19,29]. In dysmetabolic rodents, TLC-2716 and an analog (TLC-6665) reduced serum and liver TG and plasma cholesterol; these results were validated in humanized liver mice and a human liver organoid (HLO) model of steatohepatitis. Pharmacokinetics (PK) assessment of TLC-2716 revealed that its activity is primarily restricted to the liver and gut, thereby avoiding inhibition of peripheral RCT and reducing atherogenic risk. The preclinical safety of TLC-2716 was confirmed in mouse and non-human primate (NHP) toxicology studies, and its preliminary tolerability, efficacy and PK were demonstrated in a phase 1 trial in healthy participants. In conclusion, LXR repression by TLC-2716 improved lipid homeostasis without notable adverse effects (in preclinical studies and humans), supporting its further clinical development for the treatment of hypertriglyceridemia and associated cardiometabolic disorders.

## Results

### LXR is associated with dyslipidemia and other metabolic disorders

We performed a genome-wide association study (GWAS) in the UK Biobank (UKBB)[30,31]. As expected[32], genetic variants within *NR1H3*, but not *NR1H2* (Extended Data Fig. 1a), are associated with lipid biomarkers, such as TG, HDL-C and ApoA, glucose and plasma levels of insulin-like growth factor 1 (Fig. 1a). In two other unrelated human populations (FinnGen[33] and the Million Veteran Program[34]), GWAS results confirmed the associations between genetic variants within *NR1H3*, but not *NR1H2*, and its target genes (for example, *APOC3*, *APOE*, *IDOL* and *ACACA*) and metabolism-related phenotypic traits, including hyperlipidemia and plasma TG (Extended Data Fig. 1b,c). Fine-mapping indicated that rs61731956, an *NR1H3* locus, might be the potential causal signal for

HDL-C and ApoA. Burden testing also suggested a link between *NR1H3* function and lipid metabolism (Extended Data Fig. 2a,b).

We also analyzed human liver gene expression data from Gene Expression Omnibus[35], Genotype–Tissue Expression (GTEx)[36] and the Human Liver Cohort[37]. Consistent with previous studies, expression of *NR1H3* correlated positively with genes associated with amino acid, fatty acid, TG and cholesterol metabolism, indicating its essential role in lipid regulation (Fig. 1b,c). However, associations between lipid metabolism and *NR1H2* were less robust (Extended Data Fig. 1d,e). We then explored two human MASLD datasets[38,39] and found that hepatic expression of *NR1H3* is increased in individuals with MASLD Activity Scores (NAS) from 1 to 7, compared with those without MASLD (NAS = 0; Fig. 1d). Moreover, half of LXR target genes are upregulated in individuals with advanced hepatic fibrosis (stage > 2; Fig. 1e). Consistently, hepatic expression of LXR target genes, such as *Fasn*, *Scd1* and *Acaca*, positively correlates with liver steatosis and liver TG in MASLD mouse models, whereas *Srebf1* expression positively correlates with plasma TG[40] (Extended Data Fig. 2c). In addition, we performed Mendelian randomization to test whether changes in *NR1H3* expression are causally linked to lipid homeostasis. We confirmed that higher *NR1H3* expression in blood causes an increase in TG in humans, and its expression is also associated with HDL-C and liver disease markers (Fig. 1f).

### LXR inverse agonists lower TG and cholesterol in dysmetabolic animal models

TLC-2716 (ref. 41) and its analog TLC-6665 (ref. 41) are two LXR inverse agonists. Information on these compounds is available in patent US11970484B2 (ref. 41; Supplementary Fig. 1). Their potencies against LXRα and LXRβ were evaluated in biochemical binding assays, cellular mammalian two-hybrid interaction assays evaluating nuclear receptor co-repressor (NCOR) recruitment and cellular reporter assays evaluating transcriptional activity (Fig. 2a). TLC-2716 and TLC-6665 demonstrated comparable binding to and repression of LXRα and LXRβ activity with half-maximal effective concentrations (EC$_{50}$) of 7–15 and 14–17 nM, respectively (Fig. 2a). Repression of LXR activity by TLC-2716 dose dependently reduced intracellular TG accumulation in primary human Upcyte hepatocytes with an EC$_{50}$ of 289 ± 34 nM via inhibition of DNL[19] (Fig. 2b).

To evaluate the therapeutic efficacy of LXR inverse agonists in vivo, we used three different dysmetabolic rodent models, including high-fat diet (HFD)-induced obese (DIO) mice, HFD-fed Zucker diabetic fatty (ZDF) rats and HFD-fed Sprague–Dawley (SD) rats (Extended Data Fig. 3a). In these models, TLC-2716 dose dependently reduced hepatic expression of DNL genes and plasma and liver TG (Fig. 2c–e and Extended Data Fig. 3b). The area under curve (AUC) for plasma TG and total cholesterol (TC) was significantly reduced at the highest dose of TLC-2716 (1 mg per kg (body weight) per day) in DIO mice and ZDF rats (Fig. 2e,f and Extended Data Fig. 3b,c). Importantly, TLC-2716 did not impact liver injury biomarkers, including plasma alanine transaminase (Extended Data Fig. 3d) or aspartate aminotransferase (Extended Data Fig. 3e). Similarly, TLC-6665 also inhibited hepatic expression of DNL genes (Extended Data Fig. 4a–d) and reduced liver and plasma TG levels in DIO mice and ZDF rats (Fig. 2g,h).

In addition to direct effects on hepatic lipid synthesis, LXR repression exerts pleiotropic effects to reduce dyslipidemia. TLC-6665 reduced intestinal lipid absorption, measured by the uptake of ³H-labeled triolein tracer, in DIO mice (Fig. 2i). Further, TLC-2716 dose dependently inhibited ileal *Srebp1c* expression in DIO mice (Fig. 2j). In ZDF rats, TLC-2716 reduced hepatic *Angptl3* expression and plasma levels of ANGPTL3 (Fig. 2k). Similarly, liver expression of *Angptl3*, *Angptl4* and *Angptl8*, as well as *Apoc1*, *Apoc2* and *Apoc3*, was suppressed by TLC-6665 in ZDF rats (Fig. 2l). Together, these data confirm that LXR inverse agonists improve dyslipidemia via multiple mechanisms, including direct inhibition of lipid synthesis in the liver (Fig. 2c and Extended Data Fig. 4c,d) and intestine (Fig. 2j), reduced intestinal

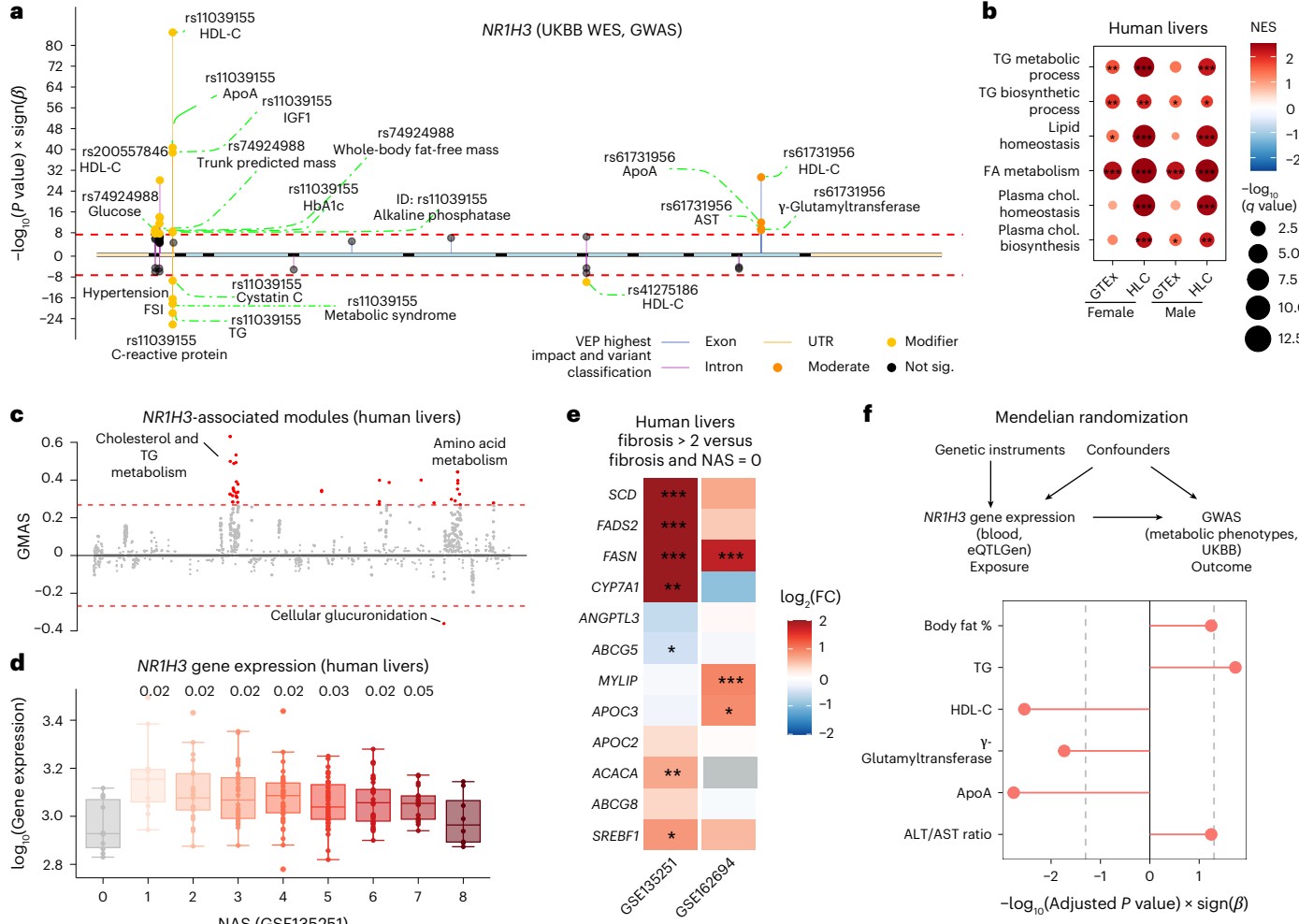

**Fig. 1 | LXR is associated with dyslipidemia and MASLD. a**, Lollipop plot showing the association between genetic variants within *NR1H3* and metabolic-related phenotypic traits based on whole-exome sequencing (WES) data in the human UKBB. The genome-wide significance threshold ($P < 5 \times 10^{-8}$) is indicated by the red dashed lines. FSI, Framingham Steatosis Index[62]; IGF1, insulin-like growth factor 1; HbA1c, hemoglobin A1c; AST, aspartate aminotransferase; sig., significant; VEP, variant effect prediction. **b**, Gene set enrichment analysis highlighting the coexpressed gene sets of *NR1H3* in both sexes across two human liver datasets (GTEx and the Human Liver Cohort (HLC)). NES, normalized enrichment score; chol., cholesterol; FA, fatty acid; *$q < 0.05$, **$q < 0.01$ and ***$q < 0.001$. **c**, Manhattan plot displaying the gene modules associated with *NR1H3* expression in human livers. Absolute gene module association score (GMAS) significance threshold (red dashed line), $|\text{GMAS}| \geq 0.268$ (ref. 35). **d**, Box plot showing the difference in *NR1H3* gene

expression in human livers according to the NAS. A two-tailed Mann–Whitney *U*-test and Benjamini–Hochberg (BH) adjustment were applied to assess the significance between individuals with MASLD (NAS score of >0) and individuals with a NAS score of 0. Adjusted *P* values are indicated; *N* = 216 individuals in total. Box plots indicate the median (center line), interquartile range (IQR; box bounds, 25th and 75th percentiles), and smallest and largest values within 1.5× IQR (whiskers). **e**, Heat map showing the association between hepatic LXR target gene expression and fibrosis scores in two different human MASH datasets; log$_2$(fold change) (log$_2$(FC)) is represented by color. The *P* value is two sided, and BH-adjusted *P* values are indicated as follows: *$P < 0.05$, **$P < 0.01$ and ***$P < 0.001$. **f**, Mendelian randomization analysis showing the causal effect of the expression of *NR1H3* in blood on plasma lipid-related phenotypic traits. ALT, alanine aminotransferase. The *P* value is two sided, and associations with a BH-adjusted *P* value of <0.1 are shown.

absorption of dietary lipids (Fig. 2i) and increased lipid clearance from circulation (Fig. 2k,l).

## LXR inverse agonists mitigate dyslipidemia-related metabolic disorders

The therapeutic effects of our LXR inverse agonists in MASLD were evaluated in the choline-deficient HFD and sodium nitrite rat model[42] (Extended Data Fig. 5a). Over 6 weeks of treatment, TLC-6665 decreased liver fibrosis (assessed histologically with picrosirius red staining; Extended Data Fig. 5b) and hepatic hydroxyproline and collagen content by 64% and 50%, respectively (Extended Data Fig. 5c,d). Liver TG (Extended Data Fig. 5e) and hepatic expression of DNL-related genes, stellate cell activation genes and fibrogenesis-related genes (*Timp1*) were also reduced (Extended Data Fig. 5f).

The effect of LXR repression by TLC-6665 (5 mg per kg (body weight) per day) on insulin sensitivity in DIO mice was assessed after 4 weeks of treatment using a two-step hyperinsulinemic–euglycemic clamp, where pioglitazone (30 mg per kg (body weight) per day) served as a positive control. The glucose infusion rate was higher in mice on TLC-6665 than in those treated with vehicle and was noninferior to pioglitazone, indicating enhanced systemic insulin sensitivity (Extended Data Fig. 5g,h). Hepatic glucose production suppression with TLC-6665 trended to be similarly reduced to pioglitazone during the 8 mU kg$^{-1}$ min$^{-1}$ insulin infusion (Extended Data Fig. 5h). Glucose utilization by muscle was also higher with TLC-6665, particularly in the oxidative soleus muscle (Extended Data Fig. 5i). A higher dose of TLC-2716 (15 mg per kg (body weight) per day) also improved glucose homeostasis in ZDF rats, as evidenced by an approximately 56%

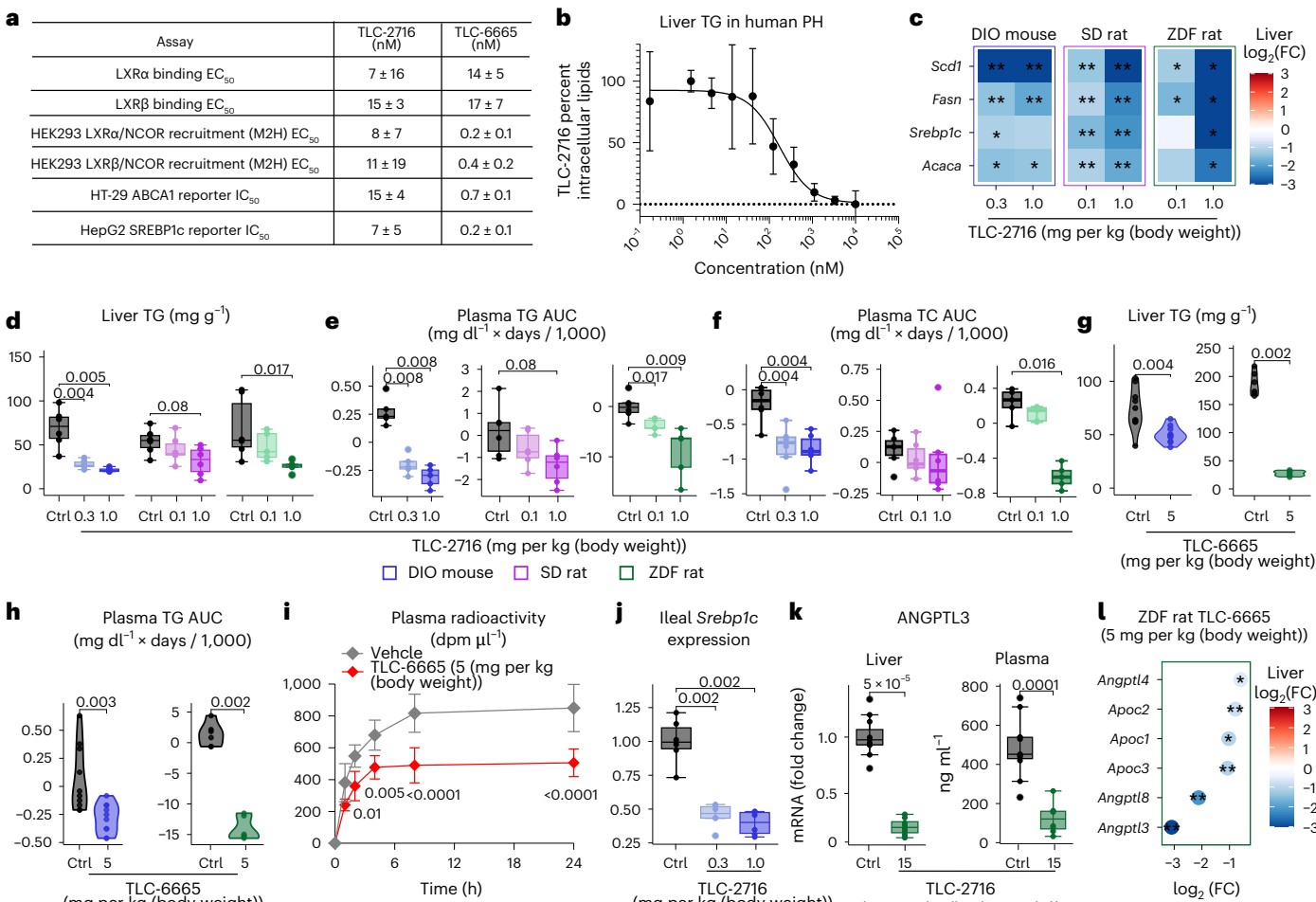

**Fig. 2 | LXR inverse agonists improve lipid homeostasis in dysmetabolic rodent models. a**, Biochemical and cellular potency of TLC-2716 and its analog TLC-6665. $IC_{50}$, half-maximal inhibitory concentration; M2H, mammalian two-hybrid. **b**, TLC-2716 lowers lipid droplet accumulation in primary human Upcyte hepatocytes (PH). Data are shown as mean ± s.d.; $N = 8$ experiments. **c**–**f**, Liver target engagement assessed by the expression of LXR target genes (**c**), liver TG content (**d**), plasma TG AUC (**e**) and plasma TC AUC (**f**) in HFD-DIO mice ($n = 6$ per group), HFD-fed SD rats ($n = 6$ per group) and HFD-fed ZDF rats ($n = 6$ for vehicle and 5 for the treatment group) treated with TLC-2716; Ctrl, control. **g**,**h**, Liver TG content (**g**) and plasma TG AUC (**h**) in DIO mice ($n = 10$ per group) and ZDF rats ($n = 6$ per group) after TLC-6665 treatment. **i**, Intestinal lipid absorption, as measured by the appearance of [$^3$H]triolein in the plasma of DIO mice treated with vehicle or TLC-6665 (5 mg per kg (body weight)) by oral gavage; $n = 5$ per

group). Data are shown as mean ± s.d.; dpm, disintegrations per minute. **j**, Ileal *Srebp1c* expression in DIO mice treated with TLC-2716 ($n = 6$ per group). **k**, Hepatic expression (left) and plasma levels (right) of ANGPTL3 in HFD-fed ZDF rats treated with vehicle or TLC-2716 for 4 weeks; $n = 11$ for vehicle and 8 for the treatment group. **l**, Hepatic transcript levels of enzymes involved in circulating TG clearance and/or hepatic TG secretion in HFD-fed ZDF rats treated with vehicle or TLC-6665; $n = 6$ per group; *$P < 0.05$, **$P < 0.01$ and ***$P < 0.001$. Data were analyzed by two-tailed Mann–Whitney $U$-test and BH adjustment (**c**–**f** and **j**–**l**), two-tailed Mann–Whitney $U$-test (**g** and **h**) or two-way analysis of variance with Sidak's multiple comparisons test (**i**). Box plots indicate the median (center line), IQR (box bounds, 25th and 75th percentiles), and smallest and largest values within 1.5× IQR (whiskers; **d**–**f**,**j** and **k**).

reduction in fasting plasma glucose after four weeks of treatment (Extended Data Fig. 5j). LXR inverse agonists hence improve insulin sensitivity in dysmetabolic rodents.

### Confirmation of treatment effects of TLC-2716 in humanized experimental models

Because cholesterol metabolism differs between rodents and humans, humanized liver chimeric PXB mice were used to confirm the effects of TLC-2716 on lipid metabolism[43]. TLC-2716 trended to reduce hepatic TG, despite the short duration (8 days) of dosing (Extended Data Fig. 6a,b). Moreover, hepatic expression of genes involved in cholesterol and TG metabolism was suggestively reduced (adjusted $P < 0.1$) with TLC-2716 (Extended Data Fig. 6c). Specifically, TLC-2716 reduced the expression of *HMGCR*, indicative of reduced cholesterol synthesis, and genes involved in hepatic LDL-C uptake (*PCSK9* and *IDOL*) and bile acids synthesis (*CYP7A1*). Reductions in the expression of *ANGPTL3* and DNL-associated genes were also observed with TLC-2716 (Extended Data Fig. 6c).

The effect of TLC-2716 on lipid accumulation was also evaluated in induced pluripotent stem (iPS) cell-derived HLOs established from human donors with different genetic backgrounds, including those with a known risk variant for metabolic dysfunction-associated steatohepatitis (MASH; the glucokinase regulatory protein (*GCKR*) rs1260326: C (GCKR^CC) > T variant (GCKR^TT))[44]. These HLOs were exposed to high concentrations of oleate for 3 days to induce a MASH-like phenotype (steatotic HLOs (sHLOs)) and then treated with TLC-2716 or vehicle (Fig. 3a). TLC-2716 dose dependently reduced intracellular lipid content in sHLOs (Fig. 3b,c). RNA sequencing revealed that sHLOs with different *GCKR* genotypes were segregated by principal component 1 (PC1), whereas PC2 separated sHLOs by TLC-2716 (5 µM; Fig. 3d). Gene set enrichment analysis further illustrated that TLC-2716 suppressed the expression of LXR-associated genes (Fig. 3e) and the expression of genes involved in lipid metabolism, inflammation and fibrosis, with a more pronounced effect in the GCKR^TT sHLOs, wherein lipid biosynthesis is upregulated[45,46] (Fig. 3f).

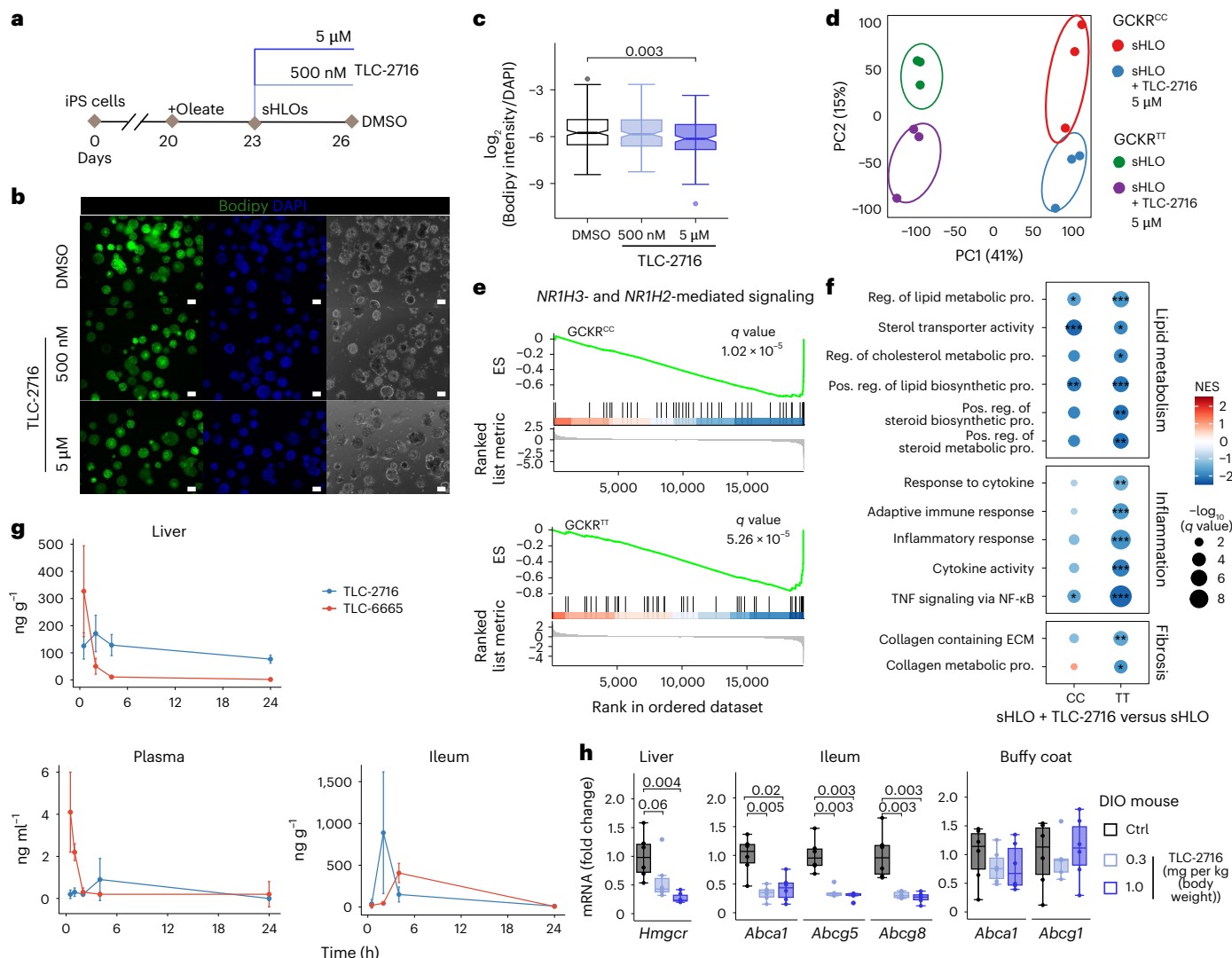

**Fig. 3 | TLC-2716 maintains lipid homeostasis in sHLOs and has a favorable therapeutic index mediated by liver and gut restriction. a**, Schematic showing the experimental pipeline of studies in sHLOs. **b**, Representative images of lipid accumulation in sHLOs. Scale bars, 100 μm. Experiments were repeated four times, and similar results were observed. **c**, Dose-dependent reduction in lipid accumulation in sHLOs treated with TLC-2716 for 3 days. *P* values were calculated by two-tailed Mann–Whitney *U*-test; *n* = 238 (DMSO), 196 (500 nM) and 153 (5 μM). **d**, PC analysis of normalized gene expression of GCKR^TT and GCKR^CC sHLOs ±5 μM TLC-2716 (*n* = 3 per group). **e**, Effects of TLC-2716 on LXR-related gene sets in GCKR^TT and GCKR^CC sHLOs (*n* = 3 per group); ES, enrichment score. **f**, Gene set enrichment analysis indicating the altered lipid-, inflammation- and fibrosis-related gene sets following TLC-2716 exposure. The effect is more robust in the GCKR^TT variant. Reg., regulation; pos., positive; pro., process; ECM, extracellular matrix. *q* values are indicated by **q* < 0.05, ***q* < 0.01 and ****q* < 0.001. **g**, Liver, plasma and ileum exposures in mice dosed with 1 mg per kg (body weight) TLC-2716 or TLC-6665 by oral gavage on day 1. Data are shown as mean ± s.d.; *n* = 3 per group. **h**, Expression of genes involved in cholesterol synthesis, metabolism and efflux/transport in liver, ileum and buffy coat in DIO mice dosed with TLC-2716 (0.3 and 1 mg per kg (body weight)) for 3 weeks (*n* = 6 per group). Significance was calculated using a two-tailed Mann–Whitney *U*-test and adjusted by the BH method. Box plots indicate the median (center line), IQR (box bounds, 25th and 75th percentiles), and smallest and largest values within 1.5× IQR (whiskers; **c** and **h**).

## TLC-2716 has a favorable therapeutic index mediated by liver and gut restriction

Plasma and tissue (liver and ileum) concentrations of TLC-2716 and TLC-6665 were evaluated in male DIO mice after 14 days of treatment. The compounds demonstrated different pharmacology in vivo, with TLC-2716 having lower plasma exposures and higher and sustained ileum and liver exposures (due to active uptake into hepatocytes) than TLC-6665 (ratio of AUCs of TLC-2716/TLC-6665 in plasma and liver of 0.17 and 6.24, respectively; Fig. 3g). TLC-2716 also dose dependently reduced the expression of transcripts involved in cholesterol synthesis (*Hmgcr*) in the liver and cholesterol efflux transporters (*Abca1*, *Abcg5* and *Abcg8*) in the intestine (Fig. 3h). However, TLC-2716 did not alter the expression of the cholesterol efflux transporters *Abca1* and *Abcg1* in the buffy coat, the fraction of blood that primarily contains white blood cells (Fig. 3h). These data indicate that TLC-2716 acts primarily in the liver and intestine and does not inhibit LXR activity in white blood cells, which are responsible for RCT and associated with atherogenic risk. Given its lower plasma exposure, TLC-2716 was selected for clinical evaluation in the treatment of severe dyslipidemias and related disorders.

The safety of TLC-2716 was assessed in preclinical toxicology studies in mice and NHPs. TLC-2716 was administered orally at 1, 5 and 15 mg per kg (body weight) per day (2- to 28-fold the maximum clinical dose based on allometric scaling) for 28 days to young, lean male and female NHPs. Dose-dependent reductions in plasma TG and TC were observed in TLC-2716-treated monkeys (Extended Data Fig. 7a,b),

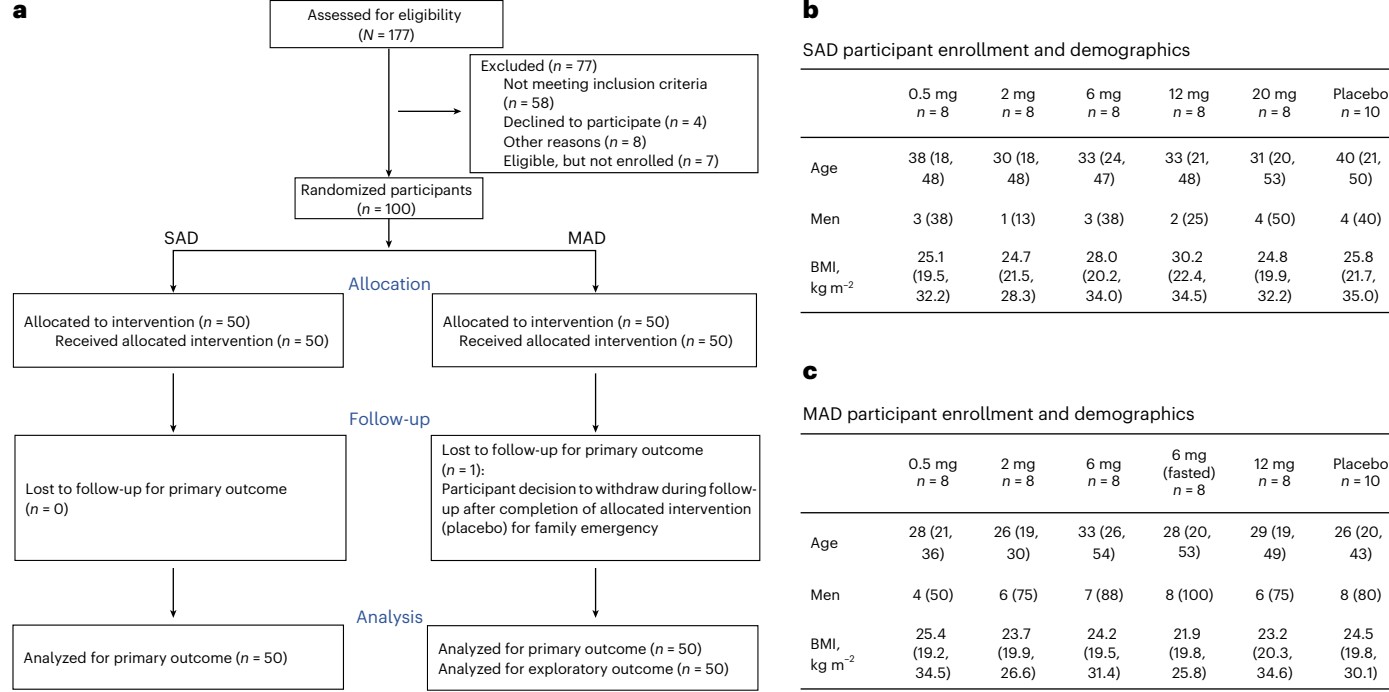

**Fig. 4 | Baseline characteristics and CONSORT flow diagram for phase 1 clinical trial. a**, Flow diagram for the CONSORT of the TLC-2716 phase 1 clinical trial. **b,c**, Baseline characterization of healthy participants enrolled in SAD and MAD cohorts in the phase 1 trial. Data are shown as median (minimum, maximum) or *n* (%).

without adverse clinical or histopathological observations at any tested doses. In mice, TLC-2716 was administered orally at 15, 60 and 120 mg per kg (body weight) per day (7- to 57-fold the maximum clinical dose) to lean male and female CD-1 mice for 26 weeks. Consistent with observations in dysmetabolic rodents, plasma TG and TC were reduced at all doses of TLC-2716 in both sexes (Extended Data Fig. 7c,d). No adverse clinical observations or changes in liver biochemistry or histopathology indicative of liver injury were present at any tested doses (Extended Data Fig. 7e,f). In both toxicology studies, the no-observed-adverse-effect level was the highest dose evaluated.

### TLC-2716 was well tolerated and improved atherogenic lipid parameters in healthy individuals

Given the favorable efficacy and toxicological profile in preclinical species, a randomized, placebo-controlled phase 1 clinical trial of TLC-2716 was conducted in humans (ClinicalTrials.gov NCT05483998, from 27 July 2022 to 18 June 2023; Fig. 4a). TLC-2716 was first tested in a single-ascending dose (SAD) study in 50 healthy individuals in which single doses of up to 20 mg were well tolerated[19,29] (Fig. 4a,b). To explore the PK and effect of multiple doses on safety-related endpoints and plasma lipids and biomarkers, another 50 healthy individuals were randomized into a multiple-ascending dose (MAD) trial (Fig. 4a,c). At baseline, mean plasma TG was $102 \pm 39$ mg dl$^{-1}$ and LDL-C was $111 \pm 30$ mg dl$^{-1}$; TG $\geq 100$ mg dl$^{-1}$, RC $\geq 20$ mg dl$^{-1}$ and LDL-C $\geq 100$ mg dl$^{-1}$ were observed in 23, 20 and 30 individuals, respectively (Table 1). Participants were randomized to receive either placebo (two participants per dose) or TLC-2716 0.5, 2, 6 or 12 mg (eight participants per dose) orally once daily for 14 days. All doses were administered in a fed state except in an additional cohort of eight participants treated with 6 mg of TLC-2716 daily for 14 days after an overnight fast. PK analysis revealed a short half-life (mean $t_{1/2}$ of 1.43–1.74 h) and time to maximal plasma concentration (mean $t_{max}$ of 2.0–4.0 h) and low maximal plasma concentrations (mean $C_{max}$ of 0.95–6.37 ng ml$^{-1}$) on day 14, consistent with rapid hepatic uptake of TLC-2716 (Extended Data Fig. 8a,b).

Following 14 days of treatment, TLC-2716 caused dose-dependent improvements in atherogenic plasma lipids (exploratory outcomes),

including TG, RC, TC, non-HDL-C, LDL-C and the number of LDL and small LDL particles (Fig. 5a–g and Table 1). Although a trend to reduced ApoB was noted (Fig. 5h), decreasing HDL-C in the placebo and the highest dose of TLC-2716 were observed (Fig. 5i). Consistent with preclinical improvement in insulin sensitivity, TLC-2716 dose dependently reduced the TG/HDL-C ratio[47] and lipoprotein insulin resistance index (Lipo-IR)[48], both exploratory surrogate markers of insulin resistance (Fig. 5j,k).

The 6- and 12-mg doses of TLC-2716 had the most favorable effects on plasma lipids (Fig. 5 and Table 1). Placebo-adjusted median percentage changes in TG from day 1 to day 14 were −37.6% (95% confidence interval (95% CI), −54.0 to −19.9) with 6 mg and −38.5% (95% CI, −67.9 to −7.2) with 12 mg (Fig. 5a and Table 1). Likewise, placebo-adjusted reductions in RC in the 6- and 12-mg groups were −33.5% (95% CI, −50.3 to −16.9) and −29.5% (95% CI, −59.1 to 2.6), respectively, when measured predose (Fig. 5b and Table 1) and −59.2% (95% CI, −90.2 to −7.1) and −61.0% (95% CI, −100 to −11.7), respectively, when measured postprandially 12 h postdose (Table 1). Finally, placebo-adjusted reductions in total LDL particles were robustly decreased: −20.3% (95% CI, −45.9 to −3.4) with the 6-mg dose and −36.5% (95% CI, −67.8 to −19.8) with the 12-mg dose (Fig. 5f and Table 1). Corresponding placebo-adjusted reductions in the number of small LDL particles were −52.2% (−100.1 to 9.8) and −60.8% (−121.4 to 0.1), respectively (Fig. 5g and Table 1). In exploratory analyses, reductions in TG, RC and LDL-C in participants treated with 6 or 12 mg of TLC-2716 were greatest in participants with higher baseline values of these parameters (Extended Data Fig. 8c–e). For example, in participants with baseline TG of ≥100 mg dl$^{-1}$, the placebo-adjusted reductions in TG from day 1 to day 14 were −45.2% (95% CI, −61.3 to −27.9) with 6 mg and −60.8% (95% CI, −70.4 to −37.9) with 12 mg.

In addition, both the 6- and 12-mg doses of TLC-2716 led to reductions in plasma levels of the LXR-related proteins ApoC3 and ANGPTL3 between days 1 and 14 (Extended Data Fig. 8f,g), suggesting that the benefits of TLC-2716 are driven, at least in part, by the repression of LXR target genes, upregulation of LPL and increased clearance of TG-rich lipoproteins. Effects on RCT were assessed by measuring *ABCA1* and *ABCG1* expression in peripheral blood mononuclear cells on day 14, at

**Table 1 | Changes in plasma lipid parameters in MAD cohorts of the phase 1 study of TLC-2716**

| Lipid parameter (unit) | Time point | Placebo (n=10) | TLC-2716 0.5 mg (n=8) | TLC-2716 2 mg (n=8) | TLC-2716 6 mg (n=16) | TLC-2716 12 mg (n=8) |
|---|---|---|---|---|---|---|
| Table 1 \| Changes in plasma lipid parameters in MAD cohorts of the phase 1 study of TLC-2716 | Day 1 | 106.3 (35.6) | 78.9 (30.8) | 109.6 (49.0) | 108.1 (37.7) | 102.0 (45.5) |
| | Change from day 1 to day 14 | 5.0 (−12.4, 22.4) | −14.3 (−34.9, 6.3) | −13.7 (−48.9, 21.5) | −35.8 (−46.8, −24.7) | −40.5 (−80.3, −0.7) |
| | Placebo-adjusted percentage change from day 1 to day 14[a] | — | −14.9 (−39.9, 6.2) | −3.7 (−34.3, 23.9) | −37.6 (−54.0, −19.9) | −38.5 (−67.9, −7.2) |
| | P value versus placebo[b] | — | 0.281 | 0.681 | 0.003 | 0.024 |
| RC, predose (mg dl[−1]) | Day 1 | 19.2 (5.7) | 15.0 (4.5) | 19.9 (8.4) | 19.6 (6.3) | 18.3 (7.5) |
| | Change from day 1 to day 14 | 1.3 (−1.6, 4.3) | −2.3 (−5.1, 0.5) | −2.3 (−8.6, 4.0) | −5.9 (−7.7, −4.0) | −5.5 (−12.4, 1.4) |
| | Placebo-adjusted percentage change from day 1 to day 14[a] | — | −16.8 (−38.6, 0) | −8.3 (−35.6, 22.6) | −33.5 (−50.3, −16.9) | −29.5 (−59.1, 2.6) |
| | P value versus placebo[b] | — | 0.126 | 0.837 | 0.006 | 0.148 |
| RC, 12h postdose (mg dl[−1]) | Day 1 | 24.8 (12.3) | 16.8 (4.6) | 25.9 (15.6) | 19.9 (7.6) | 18.9 (6.4) |
| | Change from day 1 to day 14 | 9.1 (0.5, 17.7) | 1.1 (−2.3, 4.6) | −2.8 (−16.4, 10.8) | −2.8 (−5.8, 0.1) | −4.1 (−8.3, −0.03) |
| | Placebo-adjusted percentage change from day 1 to day 14[a] | — | −33.2 (−77.3, 3.3) | −38.5 (−88.5, 14.4) | −59.2 (−90.2, −7.1) | −61.0 (−100, −11.7) |
| | P value versus placebo[b] | — | 0.095 | 0.157 | 0.032 | 0.032 |
| TC (mg dl[−1]) | Day 1 | 188.5 (33.8) | 172.8 (52.2) | 191.6 (22.9) | 187.5 (31.6) | 172.5 (21.1) |
| | Change from day 1 to day 14 | −5.9 (−28.6, 16.8) | −8.1 (−31.4, 15.1) | −23.6 (−43.2, −4.0) | −11.1 (−23.7, 1.4) | −33.0 (−51.6, −14.4) |
| | Placebo-adjusted percentage change from day 1 to day 14[a] | — | 2.8 (−19.5, 14.9) | −9.0 (−27.0, 6.8) | 0.2 (−17.4, 12.0) | −11.1 (−35.3, 6.3) |
| | P value versus placebo[b] | — | 0.975 | 0.702 | 0.975 | 0.048 |
| LDL-C (mg dl[−1]) | Day 1 | 114.9 (32.7) | 101.4 (44.1) | 120.5 (21.7) | 112.9 (29.9) | 99.0 (13.2) |
| | Change from day 1 to day 14 | −0.4 (−18.0, 17.1) | −2.9 (−21.9, 16.2) | −17.9 (−33.4, −2.3) | −3.7 (−14.9, 7.5) | −20.5 (−40.5, −0.5) |
| | Placebo-adjusted percentage change from day 1 to day 14[a] | — | 3.6 (−23.1, 22.7) | −10.5 (−37.2, 6.8) | 2.2 (−16.5, 19.0) | −17.3 (−40.1, 1.8) |
| | P value versus placebo[b] | — | 0.781 | 0.598 | 0.781 | 0.352 |
| Non-HDL-C (mg dl[−1]) | Day 1 | 134.1 (34.0) | 116.4 (45.6) | 140.4 (24.1) | 132.5 (31.6) | 117.3 (10.6) |
| | Change from day 1 to day 14 | 0.9 (−19.0, 20.8) | −5.1 (−23.6, 13.3) | −20.1 (−37.6, −2.7) | −9.6 (−20.9, 1.8) | −26.0 (−40.7, −11.3) |
| | Placebo-adjusted percentage change from day 1 to day 14[a] | — | 1.3 (−29.4, 16.7) | −10.9 (−37.6, 6.6) | −2.2 (−23.0, 11.2) | −17.2 (−38.2, −8.3) |
| | P value versus placebo[b] | — | 1.000 | 0.598 | 1 | 0.07 |
| VLDL cholesterol (mg dl[−1]) | Day 1 | 19.2 (5.7) | 15.0 (4.5) | 19.9 (8.4) | 19.6 (6.3) | 18.3 (7.5) |
| | Change from day 1 to day 14 | 1.3 (−1.6, 4.3) | −2.3 (−5.1, 0.5) | −2.3 (−8.6, 4.0) | −5.9 (−7.7, −4.0) | −5.5 (−12.4, 1.4) |
| | Placebo-adjusted percentage change from day 1 to day 14[a] | — | −16.8 (−38.6, 0) | −8.3 (−35.6, 22.6) | −33.5 (−50.3, −16.9) | −29.5 (−59.1, 2.6) |
| | P value versus placebo[b] | — | 0.126 | 0.837 | 0.0058 | 0.148 |
| HDL-C (mg dl[−1]) | Day 1 | 54.4 (14.3) | 56.4 (14.8) | 51.3 (10.6) | 55.0 (14.3) | 55.2 (15.6) |
| | Change from day 1 to day 14 | −6.8 (−11.7, −1.8) | −3.0 (−10.3, 4.3) | −3.4 (−7.3, 0.4) | −1.6 (−5.8, 2.7) | −7.0 (−12.6, −1.4) |
| | Placebo-adjusted percentage change from day 1 to day 14[a] | — | 9.6 (−5.7, 19.0) | 5.6 (−4.0, 14.2) | 8.9 (−1.8, 16.3) | 1.3 (−7.4, 7.7) |
| | P value versus placebo[b] | — | 0.336 | 0.336 | 0.336 | 0.864 |
| HDL particles (nmol l[−1]) | Day 1 | 31.2 (3.9) | 32.2 (4.4) | 30.9 (4.2) | 31.9 (5.3) | 32.3 (5.7) |
| | Change from day 1 to day 14 | −2.4 (−5.1, 0.4) | −3.7 (−8.0, 0.6) | −3.2 (−6.4, −0.1) | −5.5 (−7.5, −3.4) | −7.5 (−11.5, −3.6) |
| | Placebo-adjusted percentage change from day 1 to day 14[a] | — | −4.0 (−19.0, 12.4) | −4.7 (−16.9, 11.0) | −8.6 (−19.0, 1.2) | −14.0 (−28.9, −1.6) |
| | P value versus placebo[b] | — | 0.829 | 0.829 | 0.161 | 0.062 |
| LDL particles (nmol l[−1]) | Day 1 | 1,007.8 (304.8) | 907.5 (333.4) | 1,005.0 (198.6) | 1,032.6 (317.3) | 886.3 (145.9) |
| | Change from day 1 to day 14 | 54.0 (−130.6, 238.6) | −173.8 (−337.0, −10.5) | −171.0 (−323.8, −18.2) | −201.1 (−314.2, −87.9) | −312.9 (−465.6, −160.2) |
| | Placebo-adjusted percentage change from day 1 to day 14[a] | — | −16.6 (−46.6, −0.6) | −16.1 (−45.2, 2.2) | −20.3 (−45.9, −3.4) | −36.5 (−67.8, −19.8) |
| | P value versus placebo[b] | — | 0.058 | 0.122 | 0.023 | 0.008 |

**Table 1 (continued) | Changes in plasma lipid parameters in MAD cohorts of the phase 1 study of TLC-2716**

| Lipid parameter (unit) | Time point | Placebo (n=10) | TLC-2716 0.5 mg (n=8) | TLC-2716 2 mg (n=8) | TLC-2716 6 mg (n=16) | TLC-2716 12 mg (n=8) |
|---|---|---|---|---|---|---|
| Small LDL particles (nmol l⁻¹) | Day 1 | 479.7 (209.3) | 357.9 (171.2) | 542.1 (190.4) | 526.3 (268.3) | 431.6 (210.6) |
| | Change from day 1 to day 14 | 34.0 (−148.4, 216.4) | −81.1 (−153.5, −8.7) | −105.5 (−240.7, 29.7) | −159.5 (−265.5, −53.4) | −187.1 (−391.9, 17.6) |
| | Placebo-adjusted percentage change from day 1 to day 14[a] | — | −39.7 (−82.7, 21.8) | −34.5 (−80.6 21.5) | −52.2 (−100.1, 9.8) | −60.8 (−121.4, 0.1) |
| | P value versus placebo[b] | — | 0.230 | 0.360 | 0.205 | 0.205 |
| ApoB (mg dl⁻¹) | Day 1 | 94.3 (18.6) | 84.8 (27.7) | 96.4 (14.9) | 93.2 (19.7) | 79.7 (10.1) |
| | Change from day 1 to day 14 | 0.3 (−11.8, 12.5) | −3.1 (−18.5, 12.3) | −9.9 (−25.1, 5.4) | 0.4 (−7.3, 8.0) | −10.5 (−27.6, 6.6) |
| | Placebo-adjusted percentage change from day 1 to day 14[a] | — | 1.3 (−19.2, 19.1) | −14.2 (−23.2, 11.7) | 1.0 (−12.5, 19.1) | −10.2 (−34.2, 4.1) |
| | P value versus placebo[b] | — | 0.975 | 0.816 | 0.975 | 0.816 |
| LPIR | Day 1 | 53.2 (15.9) | 37.9 (15.4) | 51.3 (22.1) | 51.8 (18.4) | 44.9 (16.4) |
| | Change from day 1 to day 14 | 1.2 (−6.7, 9.1) | −0.5 (−7.7, 6.7) | −3.8 (−13.7, 6.2) | −13.1 (−19.4, −6.7) | −10.5 (−15.1, −5.9) |
| | Placebo-adjusted percentage change from day 1 to day 14[a] | — | 1.0 (−30.4, 28.0) | −7.7 (−28.6, 30.3) | −23.8 (−49.1, −7.1) | −26.2 (−54.1, −4.2) |
| | P value versus placebo[b] | — | 0.897 | 0.897 | 0.05 | 0.05 |
| TG/HDL-C | Day 1 | 2.2 (1.2) | 1.6 (1.0) | 2.4 (1.5) | 2.2 (1.2) | 2.1 (1.4) |
| | Change from day 1 to day 14 | 0.4 (−0.04, 0.8) | −0.3 (−0.9, 0.3) | −0.3 (−1.1, 0.5) | −0.6 (−0.9, −0.3) | −0.7 (−1.6, 0.2) |
| | Placebo-adjusted percentage change from day 1 to day 14[a] | — | −17.6 (−57.0, 6.3) | −11.0 (−48.7, 20.5) | −43.5 (−68.6, −19.2) | −44.5 (−75.4, −1.6) |
| | P value versus placebo[b] | — | 0.12 | 0.536 | 0.0018 | 0.099 |

The TLC-2716 6-mg group includes participants dosed in the fasted and fed states (n=8 each). All other doses of TLC-2716 were administered in the fed state. Data are shown as mean (s.d.) or mean (95% CI). LPIR, lipoprotein insulin resistance index; TG/HDL-C, TG/HDL-C ratio. [a]Hodges–Lehmann estimators of median (95% CI). [b]P values for comparison of change from day 1 to day 14 versus placebo were calculated by two-tailed Mann–Whitney U-test and adjusted by BH adjustment.

predose (plasma TLC-2716 levels are undetectable) and 4 h postdose (approximate $t_{max}$ of TLC-2716). TLC-2716 did not reduce the expression of either of these genes (Extended Data Fig. 8h), likely due to transient and low systemic exposure of the compound attributable to active hepatic uptake.

TLC-2716 was well tolerated in this phase 1 study[19,29]. No clinically notable changes in vital signs or safety laboratory and electrocardiogram parameters, deaths, serious adverse events (AEs) or discontinuations of study medication were reported. All treatment-emergent AEs in the TLC-2716 cohorts were mild (grade 1), except for a single moderate (grade 2) AE of thrombophlebitis in the 2-mg MAD group, which was considered unrelated to treatment. The most common AEs (reported in at least three participants overall) were mild diarrhea, headache, abdominal pain, back pain and pruritus (Extended Data Fig. 9). In general, the incidences of these AEs were similar between participants treated with placebo and TLC-2716 and were not dose dependent.

## Discussion

First-generation LXR inverse agonists have demonstrated efficacy in preclinical models for the treatment of metabolic disorders, including dyslipidemia and MASLD[19,28]. TLC-2716, developed based on this generation of LXR inverse agonists, has not been evaluated in humans, nor has any other compound of its kind[19,29]. Here, we demonstrated consistent metabolic benefits of an oral, gut- and liver-restricted LXR inverse agonist, TLC-2716, in rodent, NHP and HLO models and, ultimately, in healthy human participants, supporting the potential of this approach for the treatment of cardiometabolic diseases.

As previously described[32], genetic variants within *NR1H3* are associated with lipid metabolism in human datasets, whereas such associations were not found for *NR1H2*. Similarly, hepatic expression of *NR1H3* appears to play a more prominent role in regulating lipid metabolism, likely due to its greater hepatic expression than *NR1H2* (ref. 36). These observations may partially explain why *Nr1h2*-knockout mice exhibit lower plasma TG reductions in atherosclerosis models than *Nr1h3*-knockout mice[49], further indicating the impact of LXR

activity on the regulation of DNL and lipid metabolism[12,50]. Here, we used two LXR inverse agonists, TLC-2716 and an analog TLC-6665 (ref. 41), which potently suppress both LXRα and LXRβ. Both compounds reduced serum TG while improving cholesterol metabolism by three primary mechanisms: (1) repression of hepatic *ANGPTL3* and *APOC3* expression, causing increased LPL activity and clearance of TG-rich lipoproteins; (2) repression of hepatic *SREBP1C*, mediating reduced DNL and VLDL synthesis and release by the liver; and (3) reduced fatty acid absorption and chylomicron assembly in the intestine.

Homeostasis of LXR activity plays an important role in maintaining global metabolic health. On the one hand, the LXR pathway is upregulated in metabolic diseases, including hypertriglyceridemia and MASLD (Fig. 1d), wherein exaggerated DNL leads to hepatic steatosis, lipotoxicity, inflammation and progressive fibrosis[51]. On the other hand, *Nr1h3*-knockout mice fed a high-cholesterol and/or HFD develop a MASH-like phenotype[32,52,53]. In two recent studies, knock-in mice carrying dominant-negative *Nr1h3* mutations (W441F and W441R) exhibited more severe liver injury than *Nr1h3*-knockout mice following a MASH-inducing diet[32,52]. Overall, the exacerbation of liver damage caused by *Nr1h3* loss-of-function/knockout or dominant-negative *Nr1h3* mutations in MASH may be due to severely perturbed LXR activity in these settings. By contrast, LXR inverse agonists, which reduce the basal activity of this constitutively active receptor (favoring interaction with the co-repressor NCOR1), have demonstrated potential in reducing plasma lipids[25] and hepatic fibrosis[28]. Here, two LXR inverse agonists reduced plasma TG and TC, without inducing a MASH-related phenotype in rodents or NHPs. In a MASH rat model, 6 weeks of LXR inverse agonist treatment reduced hepatic steatosis and fibrosis (Extended Data Fig. 5b–e), and, importantly, liver biochemistry remained stable with both short-term (2–3 weeks) and long-term (26 weeks) treatment (Extended Data Figs. 3d,e and 7e,f). RNA-sequencing analysis of sHLOs further confirmed that TLC-2716 does not elevate the expression of proinflammatory or profibrotic gene expression (Extended Data Fig. 10). Indeed, consistent with alleviation of the MASH

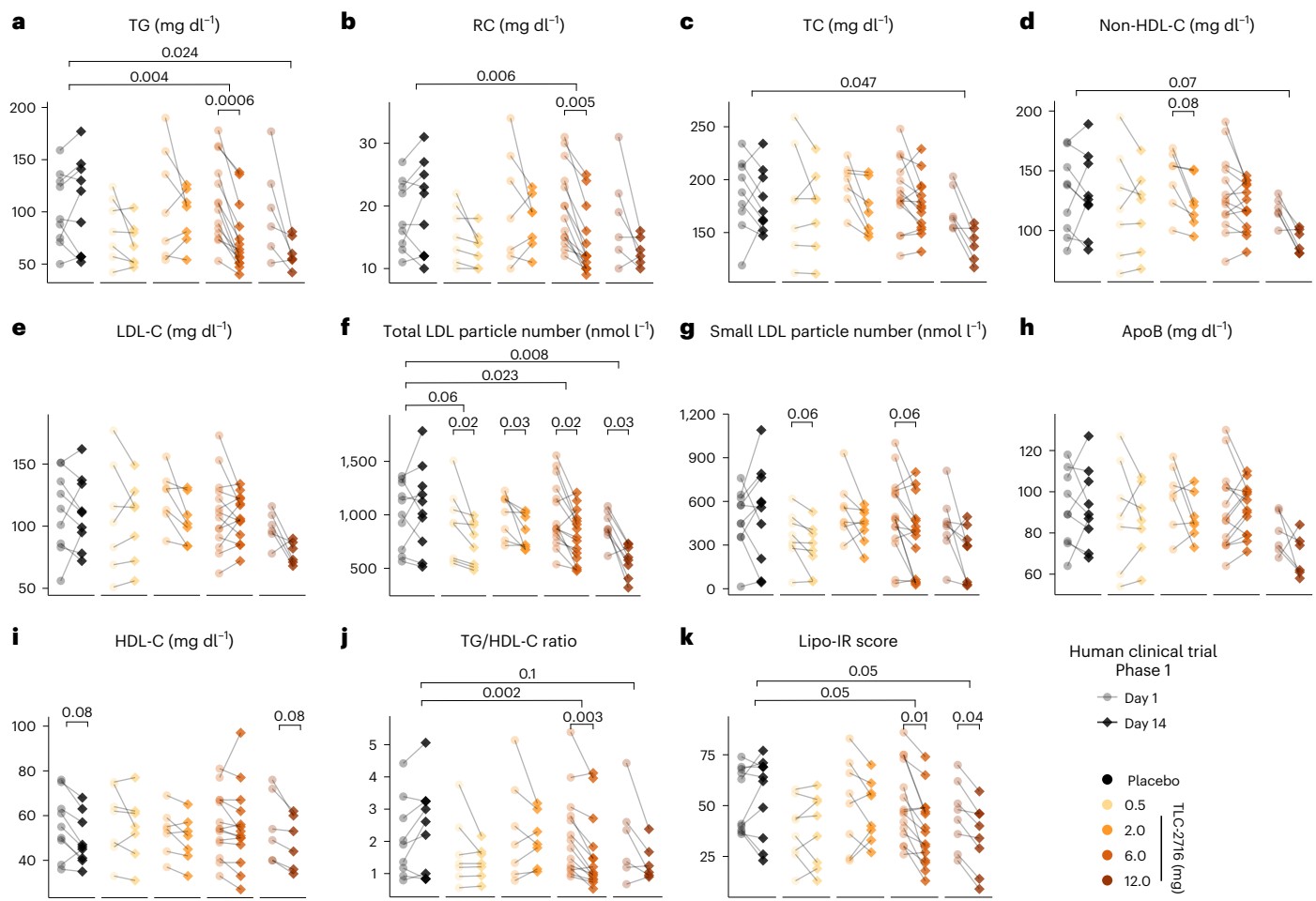

**Fig. 5 | Evaluation of the effect of TLC-2716 on lipid metabolism in humans.** **a**–**k**, Dot plots showing the therapeutic effect of 14 days of treatment with TLC-2716 at 0.5, 2, 6 or 12 mg on TG (**a**), RC (**b**), TC (**c**), non-HDL-C (**d**), LDL-C (**e**), the number of total LDL particles (**f**) and small LDL particles (**g**), ApoB (**h**), HDL-C (**i**), TG/HDL-C ratio (**j**) and Lipo-IR (**k**). The significance of comparisons between day 14 and day 1 within the same dose group was calculated by two-tailed Wilcoxon signed-rank test and adjusted by the BH method, while the significance of relative (%) changes of each parameter following 14 days of treatment compared with placebo calculated by [(day 14 – day 1) / day 1] was calculated by two-tailed Mann–Whitney $U$-test and adjusted by the BH method.

phenotype, the expression of fibrosis-related genes was downregulated following TLC-2716 treatment of sHLOs harboring a risk variant (GCKR$^{TT}$) associated with enhanced lipogenesis (Fig. 3f). Together, our data provide evidence that LXR inverse agonists restore basal LXR homeostasis in dysmetabolic models, leading to improved lipid metabolism and hepato-protection.

Dyslipidemia commonly accompanies insulin resistance and type 2 diabetes, and ectopic lipid accumulation in liver and skeletal muscle contributes causally to the development of insulin resistance[54]. Hence, we hypothesized that LXR inverse agonism may improve insulin sensitivity by reducing dyslipidemia. Indeed, TLC-2716 improved fasting glycemia in ZDF rats (Extended Data Fig. 5j), and TLC-6665 had comparable effects on insulin sensitivity to the approved PPARγ agonist pioglitazone[55] in DIO mice (Extended Data Fig. 5g). These insulin-sensitizing effects of LXR repression may offer additional cardioprotective benefit to that attributable to improvements in circulating lipids[56]. Despite comparable therapeutic effects, TLC-2716 and TLC-6665 exhibit distinct PK profiles, with TLC-2716 showing restricted distribution to both the intestine and liver due to active uptake by hepatic transporters. The limited systemic activity of TLC-2716 likely minimizes the risk of disrupting RCT (as demonstrated by unchanged expression of relevant genes in rodents and humans) and potential proatherogenic effects that have been linked to systemic LXR repression[21]. These data provide a rationale for the evaluation of TLC-2716 to treat individuals with severe dyslipidemia and suggest that TLC-2716 may reduce residual cardiovascular risk due to benefits on both lipid and glucose metabolism.

TLC-2716 was demonstrated to be safe in preclinical toxicology studies. In addition, all doses of TLC-2716 were safe and well tolerated in this phase 1 trial involving 100 healthy volunteers. TLC-2716 treatment for 14 days led to dose-dependent improvements in plasma TG and RC and other atherogenic lipids, including non-HDL-C and both total and small LDL particle number (Fig. 5). Although these data must be interpreted cautiously due to our evaluation of healthy volunteers and multiple exploratory endpoints, we observed greater benefits of TLC-2716 in individuals with higher baseline lipids, potentially due to LXR upregulation in these individuals. For example, in individuals with baseline TG of ≥100 mg dl$^{-1}$ or RC of ≥20 mg dl$^{-1}$ (roughly one-half of included participants), TLC-2716 at 12 mg led to placebo-adjusted reductions in TG and RC of >60% after only 14 days of dosing. These improvements are within the range of other therapies that target TG-rich lipoproteins, including injectable inhibitors of ApoC3 (refs. 57,58), ANGPTL3 (ref. 59) and ANGPTL4 (ref. 60), and support the potential of TLC-2716 to provide meaningful clinical benefit in individuals at risk for ASCVD. Inhibitors of ApoC3 and ANGPTL3 increase LDL-C concentrations, likely due to LPL-mediated conversion of TG-rich lipoproteins into LDL particles[61]. By contrast, the repression of LXR activity with TLC-2716 led to dose-dependent reductions in plasma LDL particles. This distinction, which is likely relevant from an ASCVD

risk perspective, presumably reflects the pleiotropic mechanisms of LXR repression, including reductions in both ANGPTL3 and ApoC3 (as demonstrated in this study), decreased hepatic synthesis of TG-rich lipoproteins and reduced intestinal lipid absorption. Overall, the oral administration of TLC-2716 may offer a compelling advantage over these alternative therapies due to patient convenience, reduced cost and the potential to combine with other lipid-lowering therapies in fixed-dose oral combinations.

Several additional findings of this phase 1 study warrant discussion. First, TLC-2716 caused dose-dependent reductions in the TG/HDL-C ratio and Lipo-IR index, both surrogate markers of insulin resistance, suggesting that TLC-2716 may improve insulin sensitivity, as observed in preclinical studies. However, because our study population comprised insulin-sensitive healthy volunteers, the potential benefits of TLC-2716 should be confirmed in insulin-resistant individuals with dyslipidemia. Similarly, because baseline lipids were relatively normal in these healthy volunteers and lipid-lowering therapies were contraindicated in this study, the benefits of TLC-2716 when added to background therapies for hypertriglyceridemia (for example, fibrates and statins) in individuals with disease require confirmation. Ultimately, these questions will be answered in upcoming phase 2 studies, including an ongoing phase 2a study of TLC-2716 in individuals with overweight/obesity with moderate-to-severe hypertriglyceridemia (TG ≥ 350 mg dl$^{-1}$) and MASLD (NCT06564584).

In conclusion, this randomized, placebo-controlled phase 1 study has demonstrated that TLC-2716 (a liver- and gut-restricted oral LXR inverse agonist) is safe and well tolerated and produces substantial improvements in plasma lipid metabolism. These clinical results are consistent with findings from genetic analysis and preclinical studies, reinforcing both safety and therapeutic promise of LXR repression in improving circulating lipids and addressing hepatic consequences of metabolic dysfunction.

## Online content

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

[1]Laboratory of Integrative Systems Physiology, Institute of Bioengineering, École Polytechnique Fédérale de Lausanne, Lausanne, Switzerland. [2]OrsoBio, Menlo Park, CA, USA. [3]Division of Gastroenterology, Hepatology and Nutrition, Cincinnati Children's Hospital Medical Center, Cincinnati, OH, USA. [4]Division of Developmental Biology, Cincinnati Children's Hospital Medical Center, Cincinnati, OH, USA. [5]WM Therapeutics, Heidelberg, Germany. [6]New Zealand Clinical Research, University of Auckland, Auckland, New Zealand. [7]Premium Research Institute for Human Metaverse Medicine (WPI-PRIMe), and Division of Stem Cell and Organoid Medicine, The University of Osaka, Suita, Japan. ✉e-mail: mani@orsobio.com; admin.auwerx@epfl.ch

## Methods

### In vitro experiments

**Biochemical binding assay.** Biochemical binding of TLC-2716 and TLC-6665 to LXRα or LXRβ was determined by time-resolved fluorescence resonance energy transfer-based biochemical assays evaluating the concentration-dependent displacement of a N-terminally biotinylated coactivator NCOA3 to recombinant glutathione *S*-transferase (GST)-tagged LXRα or LXRβ ligand-binding domain (LBD). Briefly, different concentrations of TLC-2716 or TLC-6665 were incubated with an assay buffer (Tris/HCl buffer (pH 6.8) containing 240 mM KCl, 1 μg μl$^{-1}$ bovine serum albumin and 0.002% Triton X-100) containing 530 μg ml$^{-1}$ recombinant GST-tagged LXRα/LXRβ LBD, 125 pg μl$^{-1}$ GST-Tb cryptate (CisBio), 400 nM N-terminally biotinylated NCOA3 (coactivator) peptide, 1 μM 24(*S*)-25-epoxycholesterol (LXR agonist, only in the NCOA3 assay) and 2.5 ng μl$^{-1}$ Streptavidin-XL665 (CisBio) for 1 h at 4 °C, following which fluorescence was measured in a VictorX4 multiplate reader (PerkinElmer Life Science) using 340 nm as excitation and 615 and 665 nm as emission wavelengths. Assays were performed in triplicate.

**LXR GAL4 reporter transient transfection assays.** LXRα and LXRβ activity status was determined via detection of interaction with coactivator and co-repressor proteins in mammalian two-hybrid experiments. HEK293 cells were transiently transfected with full length proteins of LXRα or LXRβ and their respective LBD domains under the pCMV-AD promoter (Stratagene). The cofactors, either the coactivator SRC1 or the co-repressor NCoR, were expressed as fusions to the DNA binding domain of the yeast transcription factor GAL4. Interaction was monitored via activation of a coexpressed Firefly luciferase reporter gene under control of a promoter containing repetitive GAL4 response elements (vector pFRLuc, Stratagene). Four hours after transfection, cells were incubated with a concentration range of TLC-2716 and TLC-6665 for 16 h, after which they were lysed, and luciferase activities were measured sequentially in the same cell extract using a BMG luminometer.

**ABCA1 and SREBP1c luciferase reporter assays.** HT-29 or HepG2 cells were stably transfected with pGL4 luciferase reporter plasmid containing the *ABCA1* or *SREBP1C*, respectively, promoter region upstream of the Firefly luciferase reporter gene and incubated with a concentration range of TLC-2716 or TLC-6665 for 16 h as described above.

**Human primary hepatocyte culture.** Human primary Upcyte hepatocytes were cultivated in medium containing high concentrations of glucose (25 mM) and palmitate (100 nM) for 5 days in the presence or absence of TLC-2716, and intracellular lipids were measured by Bodipy staining.

### Dysmetabolic rodent models

**Ethics statement.** The in vivo studies were performed at Synovo in accordance with their bioethical guidelines, which are fully compliant with ethical regulations and internationally accepted principles for the care and use of laboratory animals. Animal housing facilities were maintained at 20–22 °C with 30–50% humidity. Mice were kept on a 12-h light/12-h dark cycle with ad libitum access to food and water.

**DIO mice.** Eighteen-week-old male C57BL/6 DIO mice (14 weeks on HFD, Research Diets), purchased from The Jackson Laboratory, were dosed with vehicle (5% DMSO and 0.5% Kollidon-30 in 100 mM sodium phosphate buffer) or TLC-2716 (0.3 or 1 mg per kg (body weight)) once daily by oral gavage for 3 weeks.

**ZDF rat.** Six- to 7-week-old male obese (fa/fa) Zucker rats (ZUCKER-*Lepr*$^{fa}$, Charles River Laboratories) were fed a HFD for a total of 5 weeks and dosed with vehicle (5% DMSO and 0.5% Kollidon-30 in 100 mM sodium phosphate buffer) or TLC-2716 (0.1 or 1 mg per kg (body weight))

once daily by oral gavage for the last 2 weeks. In a second study, 6- to 7-week-old male ZDF rats (ZDF-*Lepr*$^{fa}$/Crl) were prefed a high-fat, high-cholesterol diet for 2 weeks and dosed with vehicle or TLC-6665 (5 mg per kg (body weight)) once daily by oral gavage for 4 weeks.

**SD rat.** Six- to 7-week-old male SD rats (Charles River Laboratories) were fed a HFD for a total of 5 weeks and dosed with vehicle (5% DMSO and 0.5% Kollidon-30 in 100 mM sodium phosphate buffer) or TLC-2716 (0.1 or 1 mg per kg (body weight)) once daily by oral gavage for the last 3 weeks.

**Choline-deficient HFD rat.** Six- to 8-week-old male Wistar rats (Charles River Laboratories) were fed an L-amino acid rodent diet with 60 kcal% fat with no added choline and 0.1% methionine (Research Diets, A06071302) for a total of 12 weeks. After 4 weeks of diet feeding, animals started receiving sodium nitrite injections (25 mg per kg (body weight) intraperitonially) three times a week for the remainder of the study. After 6 weeks of choline-deficient HFD feeding, animals were dosed with vehicle (5% DMSO and 0.5% Kollidon-30 in 100 mM sodium phosphate buffer) or TLC-6665 (5 mg per kg (body weight)) once daily by oral gavage for the last 6 weeks.

**Humanized liver mice.** Human liver chimeric PXB mice were purchased from PheonexBio, and in-life procedures were performed at InterVivo Solution in accordance with their bioethical guidelines, which are fully compliant with ethical regulations and internationally accepted principles for the care and use of laboratory animals. Briefly, animals were assigned to two groups ($n$ = 5 male mice per group), vehicle (5% DMSO and 0.5% Kollidon-30 in 100 mM sodium phosphate buffer) or TLC-2716 (1 mg per kg (body weight) daily by oral gavage) for 8 days, after which animals were humanely killed by cardiac puncture, and plasma and tissues were collected for lipid and gene expression endpoints as described below.

### Histopathology on liver tissues

Liver tissue was dipped into ice-cold PBS for 1 min and then fixed in 4% paraformaldehyde in PBS overnight for a maximum of 16 h. Afterward, samples were transferred to cold 70% ethanol/PBS to avoid prolonged fixation and embedded in paraffin before sectioning. Sections were stained with picrosirius red to visualize collagen.

### Plasma parameter measurement

Plasma samples were analyzed for levels of alanine aminotransferase, aspartate aminotransferase, cholesterol and triglycerides using respons 910 (Diasys Diagnostics Systems), as per the manufacturer's instructions.

### Liver parameter measurements

Collected frozen liver samples were ground to a fine powder with a pestle and mortar under liquid nitrogen. An aliquot was subjected to lipid isolation using organic solvents (hexane and isopropanol), and, after extraction, triglyceride and cholesterol levels were quantified using commercially available kits (FUJIFILM Wako Chemicals Europe). An additional aliquot was used for collagen determination after acid hydrolysis (Total Collagen Assay, Quickzyme Biosciences).

### Two-step hyperinsulinemic–euglycemic clamp

The study was performed at Physiogenex S.A.S. in accordance with ethical regulations, Guide for the Care and Use of Laboratory Animals (revised 1996 and 2011, 2010/63/EU) and French laws.

Twenty-four-week-old DIO mice (Jackson Laboratories) were fed a HFD (Research Diets, D12492) for 18 weeks and dosed with vehicle (5% DMSO and 0.5% hydroxypropyl methylcellulose in phosphate-buffered saline), TLC-6665 (5 mg per kg (body weight)) or pioglitazone (30 mg per kg (body weight)) once daily by oral gavage for 4 weeks.

Animals were subjected to a two-stage hyperinsulinemic–euglycemic clamp procedure after a 6-h fast and 2 h after the last dose. Briefly, animals received a bolus of D-[3-$^3$H]glucose (30 µCi per mouse) followed by D-[3-$^3$H]glucose (30 µCi min$^{-1}$ kg$^{-1}$) infusion (2 µl min$^{-1}$) for up to 210 min. Insulin was simultaneously infused at 8 mU kg$^{-1}$ min$^{-1}$ for the first 100 min and at 18 mU kg$^{-1}$ min$^{-1}$ for the last 110 min. Blood glucose was measured from the tip of the tail every 10 min by a glucometer. The glucose infusion rate was adjusted according to blood glucose levels until a first euglycemic steady state was reached (from ~70 to 100 min of infusion) and similarly adjusted until a second euglycemic steady-state was reached (from ~150 to 210 min of infusion). During both steady states, blood (5 µl) was collected frequently from the tail tip for $^3$H-radioactivity measurements and calculations of glucose flux.

## Toxicity study

**CD-1 mice.** A 26-week Good Laboratory Practice (GLP) toxicology study was conducted at Inotiv according to the protocol, Inotiv's Standard Operating Procedures and in compliance with the current US Food and Drug Administration GLP Regulations for Non-Clinical Studies (21 CFR Part 58). Animal housing facilities were maintained at 20–26 °C with 30–70% humidity. Mice were kept on a 12-h light/12-h dark cycle with ad libitum access to food and water. Briefly, the safety profile of TLC-2716 was evaluated in CD-1 mice (Charles River Laboratories) administered orally once daily at 0 (vehicle: 100 mM sodium phosphate puffer, 0.5% polyvinylpyrrolidone (PVP; pH 7.4) and 3% DMSO), 15, 60 and 120 mg per kg (body weight) per day for 26 weeks. This terminal study included comprehensive assessments of toxicokinetics and clinical laboratory parameters (hematology, clinical chemistry and urinalysis). Animals were monitored daily for clinical signs, with regular measurements of body weight, food consumption and vital signs. Necropsy and detailed tissue analysis were performed at study termination.

**NHPs.** A 4-week GLP toxicology study was conducted in cynomolgus monkeys (Guangzhou Xiangguan Biotech) to evaluate the safety profile of TLC-2716 administered orally once daily at 1, 5 and 15 mg per kg (body weight) per day. This terminal study included comprehensive assessments of toxicokinetics and clinical laboratory parameters (hematology, clinical chemistry and urinalysis). Animals were monitored daily for clinical signs, with regular measurements of body weight, food consumption and vital signs. Necropsy and detailed tissue analysis were performed at study termination. All procedures involving animals were reviewed and approved by the Institutional Animal Care and Use Committee and were conducted in accordance with international guidelines for the care and use of laboratory animals.

## RNA isolation for real-time quantitative PCR with reverse transcription

Target organs (liver and intestine) were subjected to gene expression analysis by real-time quantitative PCR with reverse transcription (RT–qPCR). Aliquots of liver and ileum were snap frozen in liquid nitrogen and subsequently homogenized with a pestle and mortar under liquid nitrogen. Aliquots of the homogenized tissues were processed for RNA extraction and cDNA synthesis. Aliquots from the reverse-transcribed samples were used for the detection of specific mRNA transcripts of different genes of interest by RT–qPCR using commercially available mouse cDNA sequence-specific PCR primers and Taqman probes. Resulting cycle threshold ($C_t$) values were normalized to the housekeeping gene *Tbp* measured in the same sample. Results are plotted as fold change compared with vehicle controls. All primers or product codes for RT–qPCR are indicated in Supplementary Table 1.

## HLO models of steatohepatitis

HLOs were generated as previously described[63]. Briefly, human iPS cells were differentiated into foregut progenitor cells using a published protocol[44]. To initiate organoid formation, foregut cells were resuspended in Matrigel at a final concentration of 750,000 cells per ml. Fifty-microliter droplets of the cell–Matrigel mixture were plated and cultured in Advanced DMEM/F12 (Gibco) supplemented with B27, N2, 10 mM HEPES, 1% GlutaMAX, 1% penicillin–streptomycin, 5 ng ml$^{-1}$ FGF2, 10 ng ml$^{-1}$ VEGF, 20 ng ml$^{-1}$ EGF, 3 µM CHIR99021, 0.5 µM A83-01 and 50 µg ml$^{-1}$ ascorbic acid for 4 days. The medium was then replaced with the same basal formulation containing 2 µM retinoic acid and cultured for an additional 4 days. Finally, the organoids were maintained in hepatocyte culture medium (Lonza) for 6 days to promote hepatic maturation. To induce a steatohepatitis-like phenotype, HLOs were gently retrieved from the Matrigel and washed with PBS. HLOs were then exposed to 300 µM sodium oleate (Sigma) in hepatocyte culture medium for 3 days using ultra-low attachment six-well plates (Corning). For quantification of intracellular lipid accumulation, sHLOs were rinsed three times with prewarmed PBS and stained with 2 µM BODIPY 493/503 (Thermo Fisher Scientific) for neutral lipid detection, along with NucBlue Live ReadyProbes (Thermo Fisher Scientific) for nuclear counterstaining. Fluorescent images were acquired using a Keyence BZ-X710 automated fluorescence microscope. Lipid droplet volume was quantified using the Hybrid Cell Count application (Keyence) and normalized to nuclear signal intensity. For RNA sequencing, total RNA was extracted from iPS cell-derived HLOs using an RNeasy Plus Mini kit (Qiagen) following the manufacturer's protocol. RNA quality were assessed using a Fragment Analyzer (Advanced Analytical), and only samples with an RNA Quality Number of ≥8.0 were used for library preparation.

## *GCKR* single-nucleotide polymorphism genotyping

Genotypes of *GCKR* variants were determined using an Illumina Infinium Global Diversity Array with the Enhanced PGx platform (Illumina) according to the manufacturer's instructions. Genomic DNA was extracted from undifferentiated iPS cells using a QIAamp DNA Mini kit (Qiagen). Genotyping was performed at Cincinnati Children's Hospital Medical Center Genomics Sequencing Facility using standard Infinium HTS assay protocols. Quality control steps included removal of samples with a call rate of <98%, excess heterozygosity or sex mismatch. Single-nucleotide polymorphisms (SNPs) with a call rate of <95%, Hardy–Weinberg equilibrium $P$ of <1 × 10$^{-6}$ or minor allele frequency of <1% were excluded from downstream analysis. Genomic positions were mapped to the GRCh38 (hg38) human reference genome.

## Human genetic studies

We analyzed metabolic-related phenotypic traits from the UKBB[30,31] under Application Number 48020 by focusing on participants of European ancestry only (based on the UKBB return dataset 2442). A GWAS was performed using REGENIE. Step 1 involved the estimation of population structure using genotyping arrays (UKBB field 22418). Step 2 calculated genetic variant–phenotype associations based on the WES data (UKBB field 23159) using the following covariates: the first ten genetic PCs, age, sex and age–sex interaction. Gene-based testing (burden test) was performed using REGENIE step 2 by collapsing single-variant test statistics into gene-level results. Variants were grouped into functional sets defined by mask objects, where each mask represents a labeled variant category that combines different types of annotations.

Summary statistics from the FinnGen[33] study were downloaded from https://www.finngen.fi/en/access_results, and those for the Million Veteran Program[34] database were accessed through dbGaP under accession number phs002453.v1.p1 through the AgingX project (ID 10143). The significant ($P < 5 × 10^{-8}$) associated phenotypic traits of genetic variants within candidate genes in each human population were further extracted. Fine-mapping results were downloaded from https://www.finucanelab.org/data (ref. 64).

## Mendelian randomization analysis

Significant *cis*-eQTLs of LXRα (*NR1H3*) and LXRβ (*NR1H2*) in liver or blood were obtained from the GTEx (version 8, 208 individuals)[36] or eQTLGen (31,684 individuals)[65]. Linkage disequilibrium clumping for SNPs was estimated using Plink (genetics.binaRies R package, window size 1 Mb, minor allele frequency ≥1%) on the 1000 Genomes reference panel[66]. Only one independent SNP was found within *NR1H3* using the GTEx liver *cis*-eQTLs, whereas eight independent SNPs were identified using the eQTLGen blood *cis*-eQTLs, suggesting that the eQTL-Gen blood *cis*-eQTLs are more reliable for Mendelian randomization analysis. Therefore, significant *cis*-eQTLs of LXR in blood extracted from eQTLGen were applied as exposures in the Mendelian randomization analysis. GWAS summary statistics for plasma lipid-related traits were derived from WES data in the UKBB and served as outcomes. The causal effects of *NR1H3* and *NR1H2* gene expression on lipid traits were estimated by Inverse Variance Weighted approach using the TwoSampleMR[67–70] R package (v0.6.2).

## Coexpression gene set analysis

Liver gene expression datasets were obtained from GTEx[36] and the Human Liver Cohort[37]. Correlation coefficients between each gene and LXRα (*NR1H3*) or LXRβ (*NR1H2*) were calculated by Pearson correlation. Genes were ranked by correlation coefficient, and gene sets were extracted using the msigdbr R package (version 7.5.1)[71]. Gene set enrichment analysis was performed using the clusterProfiler R package (4.12.6)[72].

## RNA-sequencing analysis

RNA was sequenced by BGI with the DNBSEQ platform. The quality of the reads was then verified using FastQC (version 0.11.9). Low-quality reads were removed, and no trimming was needed. Alignment was performed against the human genome (GRCh38, v113) following the STAR (version 2.73a) manual guidelines[73]. Normalized effective library sizes were calculated by trimmed mean of M values. The voom function of the Limma R package (version 3.60.0)[74] was applied to transform gene counts for linear modeling with precision weights. The differential expression analysis for the transcriptome was performed using the R package Limma (version 3.60.0)[74]. Genes were ranked by $\log_2$ (fold change), and gene sets were extracted using the msigdbr R package (version 7.5.1)[71]. Gene set enrichment analysis was performed using the clusterProfiler R package (version 4.12.6)[72].

## Phase 1 clinical trial

**Study oversight.** The study protocol was approved by the Northern B Health and Disability Ethics Committee (2022 FULL 12858). This study is registered at ClinicalTrials.gov (registration: NCT05483998) and was conducted at a single site in New Zealand (Auckland Clinical Research) from 27 July 2022 to 18 June 2023, in accordance with relevant local regulatory policies. Written informed consent was obtained before enrollment.

**Study design.** The randomized, placebo-controlled phase 1 study included SAD and MAD cohorts. In the SAD cohorts, healthy participants were treated with single oral doses of TLC-2716 (0.5, 2, 6, 12 and 20 mg) or placebo, and in the MAD cohorts, participants received once-daily oral doses of TLC-2716 (0.5, 2, 6 and 12 mg) or placebo for 14 days. For each cohort, eight participants were randomized to receive TLC-2716, and two participants were randomized to receive placebo; study medications were administered in a fed state within 5 min of completing a standardized breakfast. In an additional MAD cohort, eight participants received TLC-2716 (6 mg) after an overnight fast to evaluate the effects of fasted versus fed dosing on the PKs of TLC-2716. Participants were confined in the study center until 72 h following the last dose of study drug, and all participants returned to the study center 14 days after the last dose of study drug for a safety follow-up visit. To maximize participant safety in SAD cohorts, safety and tolerability of the study drug through day 2 were assessed in two sentinel participants (one randomized to TLC-2716 and one randomized to placebo) per cohort before dosing of the remaining participants in each cohort. In addition, TLC-2716 dose escalation was preceded by a review of safety, PK and pharmacodynamic data by a safety review committee consisting of the principal investigator and sponsor representatives.

**Inclusion and exclusion criteria.** In brief, eligible study participants were healthy, nonsmoking men and women between 18 and 55 years of age and with a BMI from 19 to 35 kg m$^{-2}$, inclusive at screening. All participants had an estimated glomerular filtration rate of ≥80 ml min$^{-1}$, normal liver biochemistry (total bilirubin 1.0- to 1.5-fold the upper limit of normal was permitted in participants with Gilbert's syndrome) and 12-lead electrocardiograms and screening laboratory evaluations (for example, hematology, chemistry and urinalysis) that were normal or considered to have no clinical importance by the investigator. In the MAD cohorts, an attempt was made to enroll participants with TG of ≥150 mg dl$^{-1}$ and/or LDL-C of ≥130 mg dl$^{-1}$ to enable preliminary assessment of the lipid-lowering benefits of TLC-2716. Key exclusion criteria included women who were pregnant or lactating, TG of ≥500 mg dl$^{-1}$, LDL-C of ≥190 mg dl$^{-1}$, the presence of serious active medical or psychiatric illness, excessive alcohol consumption (defined as greater than 21 units per week for men and 14 units per week for women), substance abuse or recent receipt of an investigational compound. Participants who had taken any prescription or over-the-counter medications, including herbal products, within 28 days before the start of study drug dosing, except vitamins, acetaminophen, ibuprofen and/or hormonal contraceptives, were excluded. A complete list of inclusion and exclusion criteria is available from the authors.

## Safety, PK and pharmacodynamic assessments

**Primary outcome.** Safety and tolerability assessments, including AE monitoring, laboratory tests, physical examinations and electrocardiogram evaluations, were performed throughout the study. AEs were graded according to the Common Terminology Criteria for Adverse Events Grading Scale v5.0. Intensive PK sampling over 72 h after dosing on day 1 (in SAD and MAD cohorts) and day 14 (in MAD cohorts) was conducted, and TLC-2716 plasma concentrations were determined using a validated liquid chromatography–tandem mass spectrometry assay. PK parameters were estimated via noncompartmental methods using Pheonix WinNonlin 6.2.1 and 8.3.4 (Certara).

**Secondary outcome.** Plasma lipid parameters, including ApoB, were evaluated by NMR LipoProfile (Labcorp). RC was calculated as the difference between TC and the sum of direct LDL-C and HDL-C. Unless indicated, lipid parameters were measured before dosing following an overnight fast. Plasma ApoC3 and ANGPTL3 were evaluated by ELISA (360biolabs), and the impact of TLC-2716 on peripheral RCT was evaluated by assessing changes in the expression of *ABCA1* and *ABCG1* in peripheral blood mononuclear cells, before dosing and 4 h after dosing on day 14 (Gnomix).

**Statistical analyses.** Due to its exploratory nature, no formal power or sample size calculations were used to determine the sample size for this study. Empirically, the sample size was selected to adequately characterize the safety, PKs and pharmacodynamics of TLC-2716. Given the skewed distribution of plasma lipid data, we applied the Hodges–Lehmann estimator in deriving placebo-adjusted percentage change from baseline values in lipid parameters. This method estimates the median (and 95% confidence interval) of all possible pairwise differences between treatment groups and addresses the potential impact of skewed data or outliers. For analyses of changes from baseline, two-tailed Wilcoxon signed-rank tests (for comparisons within groups) and two-tailed Mann–Whitney *U*-tests (for comparisons

between groups) were used, which were adjusted using the BH adjustment method. In this clinical trial study, sex is self-reported. Due to the small sample size of each group, sex is not considered in the analysis.

## Reporting summary

Further information on research design is available in the Nature Portfolio Reporting Summary linked to this article.

## Data availability

The study protocol and statistical plan are available within the article and Supplementary Information. The raw data for preclinical experiments and the summary results for the clinical trial phase 1 data are available in the Source Data, whereas the individual data from the phase 1 clinical trial are available upon reasonable request from academic or qualified clinical researchers affiliated with recognized institutions, strictly for the purpose of conducting noncommercial, ethically approvable research aligned with the original scope of the trial. Applicants are required to submit a detailed research proposal, curriculum vitae and declaration of non-conflict of interest. Requests must clearly describe the research objectives and methodology and must be reviewed and approved by the corresponding authors. All approved requestors will be required to sign a data access agreement that restricts data use solely to the approved research project and prohibits any further distribution. The HLO RNA-sequencing data are available under GEO number GSE299888. Source data are provided with this paper.

## Code availability

This study did not generate original code.

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

## Acknowledgements

We thank all members of the Auwerx, Takebe and OrsoBio teams for helpful discussions. We thank Z. Kutalik (University of Lausanne) for expert guidance in statistical genetics. The work in J.A.'s group was supported by grants from the EPFL, and the work in T.T.'s group was supported by grants NIH DP2 DK128799-01, R01DK135478, PHS Grant P30 DK078392, AMED JP24gm1210012, JP24fk0210150, JP23fk0210106, JST JPMJMS2022, JPMJMS2033 and World Premier International Research Center Initiative (WPI) PRIMe, MEXT, Japan.

## Author contributions

X.L., R.P.M., G.M.S. and J.A. conceived the project. A.V., N.S., E.M. and C.K. performed all in vivo studies. R.S.H., S.W., B.J.K., E.J.G. and R.P.M. oversaw the phase 1 clinical trial. M.K. and T.T. performed the sHLO experiments. G.V.G.v.A. provided the GWAS summary statistics in UKBB. X.L., B.J.K. and R.P.M. performed data analysis. G.B., A.V., R.P.M., G.M.S. and T.T. provided scientific advice and materials. J.A. and G.M.S. supervised the study. X.L., R.P.M. and J.A. wrote the paper with contributions from all authors.

## Competing interests

A.V., R.S.H., S.W., E.M., B.J.K., R.P.M. and G.M.S. are employed by OrsoBio, and J.A. and T.T. are advisors to OrsoBio. C.K. is employed by WM Therapeutics. The other authors declare no competing interests.

## Additional information

**Extended data** is available for this paper at https://doi.org/10.1038/s41591-025-04169-6.

**Correspondence and requests for materials** should be addressed to G. Mani Subramanian or Johan Auwerx.

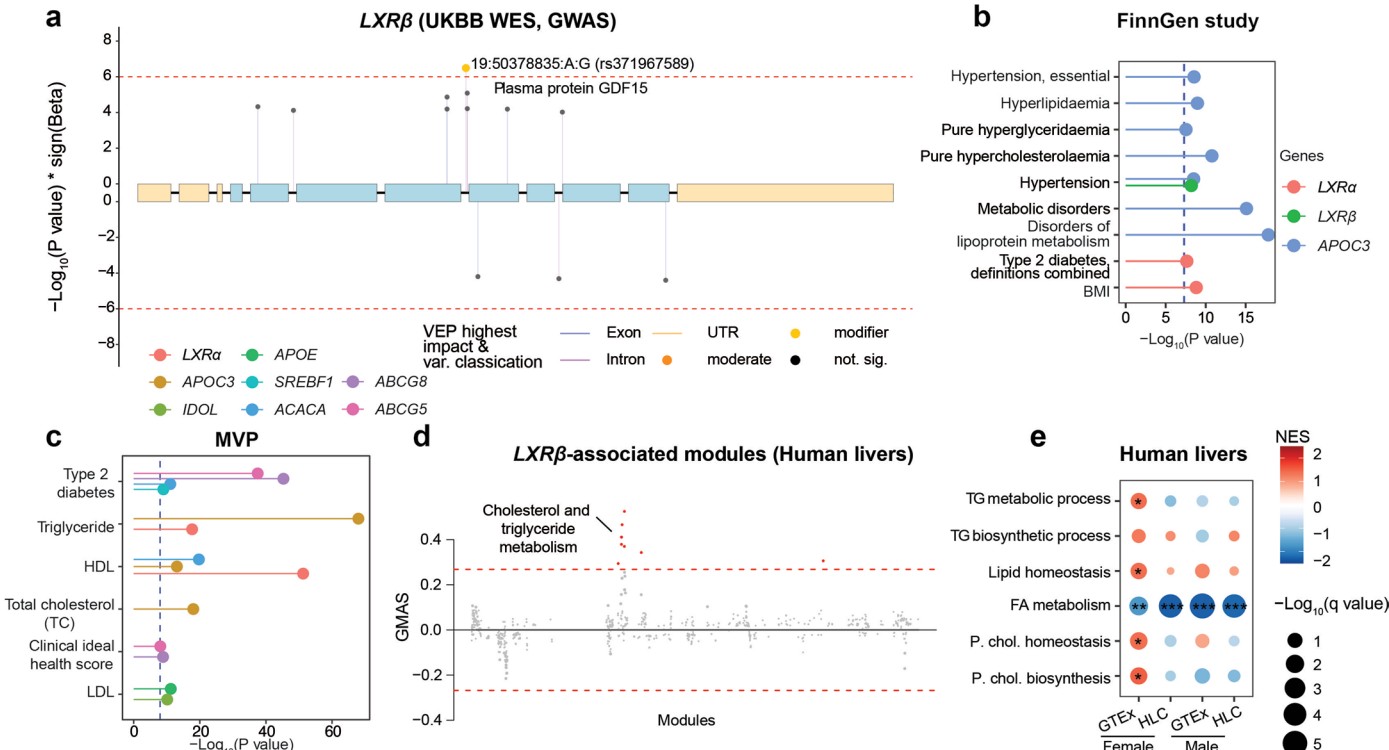

**Extended Data Fig. 1 | Associations between dyslipidemia and *LXRβ* or genes mediated by LXR activity. (a)** Lollipop plot showing the association between genetic variants within *LXRβ* based on whole exome sequencing (WES) data in the human UK Biobank (UKBB). The suggestive significance threshold is represented by the red dashed lines: -Log₁₀(P value) ≥ 6. **(b, c)** Dot plots showing the MASLD/lipid metabolism-related clinical traits that have GWAS hits within *LXRα, LXRβ* and LXR target genes in the FinnGen study **(b)** and the Million Veteran Program (MVP) **(c)**. Only significant associations (P value < 5 × 10⁻⁸) were shown.

**(d)** Manhattan plot displaying the gene modules associated with *LXRβ* expression in human livers. Absolute gene-module association score (GMAS) significance threshold (red dashed line): |GMAS| ≥ 0.268. **(e)** Gene set enrichment analysis highlighting the co-expressed genesets of *LXRβ* in both sexes across two human liver datasets (the GTEx and the Human Liver Cohort [HLC]). Significance: *q value < 0.05; **q value < 0.01; ***q value < 0.001. P: plasma; TG: triglyceride; FA: fatty acid.

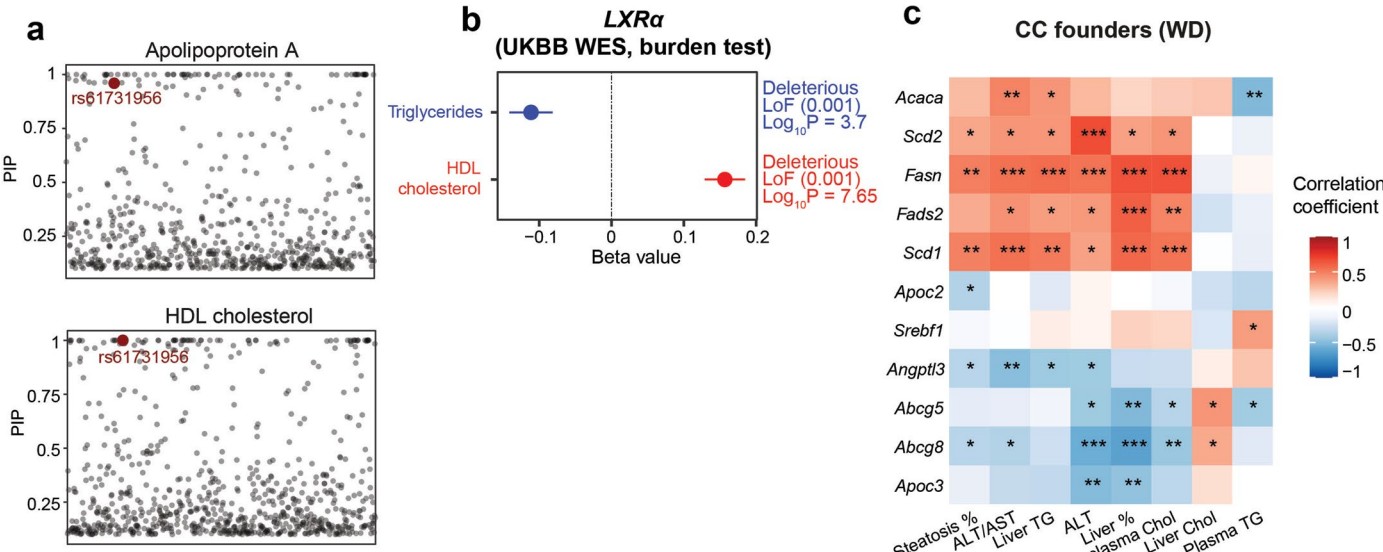

**Extended Data Fig. 2 | Associations between dyslipidemia and *LXRα* or genes regulated by LXR activity. (a)** Fine-mapping analysis showing the potential causal variants for Apolipoprotein A and HDL cholesterol calculated by SuSiE method based on human UKBB database. Only genetic variants within 95% credible sets and posterior inclusion probability (PIP) > 0.1 are shown. The genetic variant located in *LXRα* with a posterior inclusion probability (PIP) > 0.9 is highlighted in red. **(b)** Burden test using the UKBB whole exome sequence indicating the effect of loss of function (LoF) variant group on plasma lipid metabolism. Deleterious LoF variants contain LoF and missense variants. Genetic variants with minor allele frequency < 0.001 were grouped. N = 371,089

and 370,799 for European participants with HDL cholesterol and triglycerides, respectively. Data are shown as beta ± SE. The P value is two-sided. **(c)** Heatmap illustrating the Pearson correlation between hepatic gene expression of LXR-regulated genes and MASLD-related traits in collaborative cross (CC) founder mice fed western diet (WD) and housed at thermoneutrality to induce MASLD. Liver%: liver weight expressed as percentage of the body weight. Steatosis%: percentage of tissue area occupied by fat vacuoles quantified from H&E-stained liver sections. Correlation coefficients are represented by color and Benjamini-Hochberg (BH) adjusted P values are indicated as follows: *<0.05; **<0.01; ***<0.001.

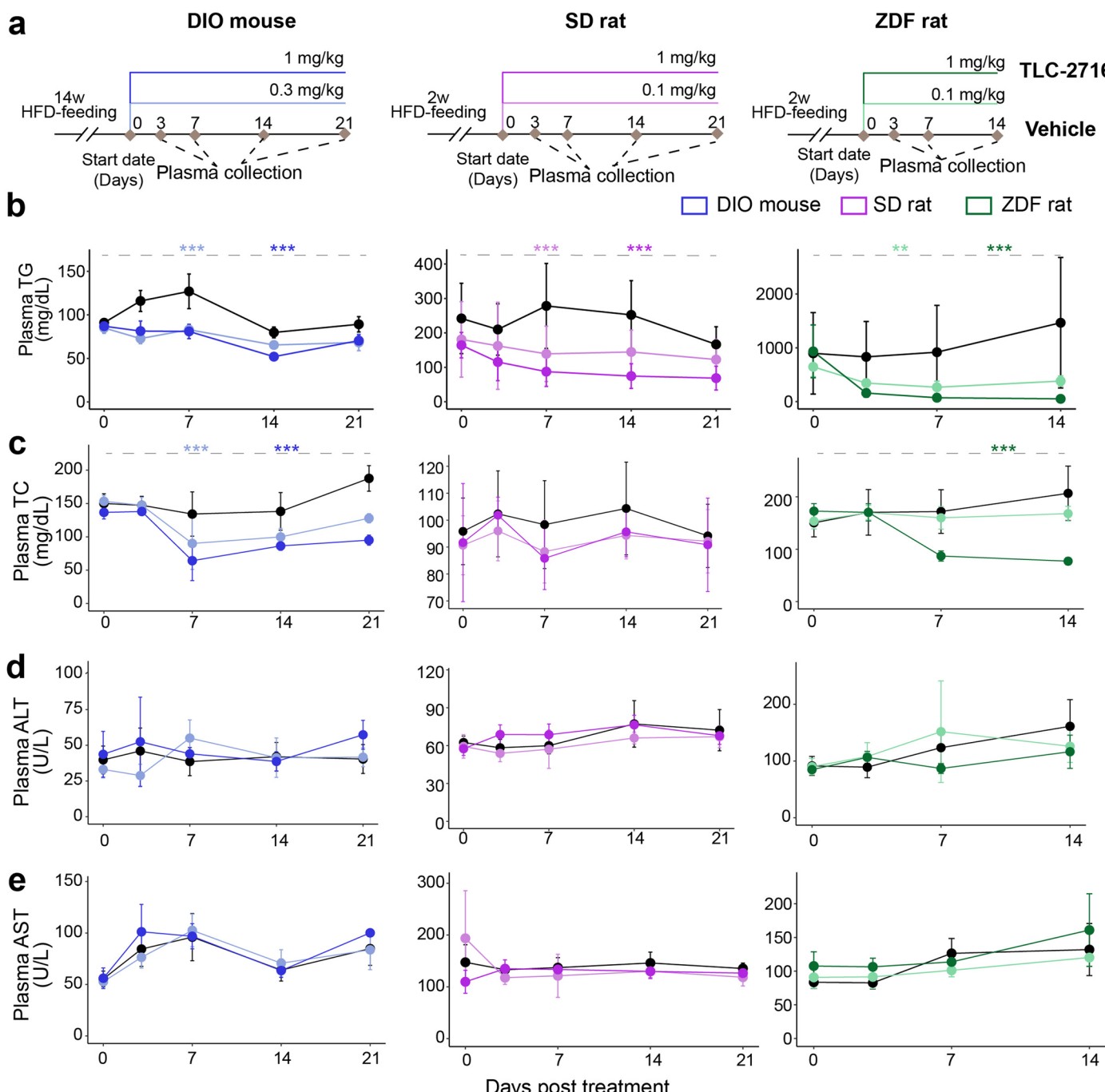

**Extended Data Fig. 3 | TLC-2716 lowers plasma total cholesterol and triglyceride without affecting liver biochemistry. (a)** Schematic illustrating the study design in diet-induced obese (DIO) mice (n = 6/group), high-fat diet (HFD)-fed Sprague-Dawley (SD) rats (n = 6/group), and HFD-fed Zucker diabetic fatty (ZDF) rats (n = 6 for vehicle and 5 for treatment group) treated with TLC-2716. **(b-e)** Plasma TG **(b)**, plasma TC **(c)**, plasma alanine aminotransferase (ALT) **(d)**, and aspartate aminotransferase (AST) levels **(e)** in DIO mice, SD rats, and ZDF

rats treated with TLC-2716 once daily by oral gavage for 14-21 days. Black lines indicate vehicle-treated animals. Significance was calculated by Two-way ANOVA with formula lipid-related parameter ~ treatment condition + days + interaction between treatment condition and days, followed by Tukey Honest Significant Differences test. Adjusted P values were indicated as follows: * <0.05; ** <0.01; *** <0.001. Data are shown as mean ± s.d.

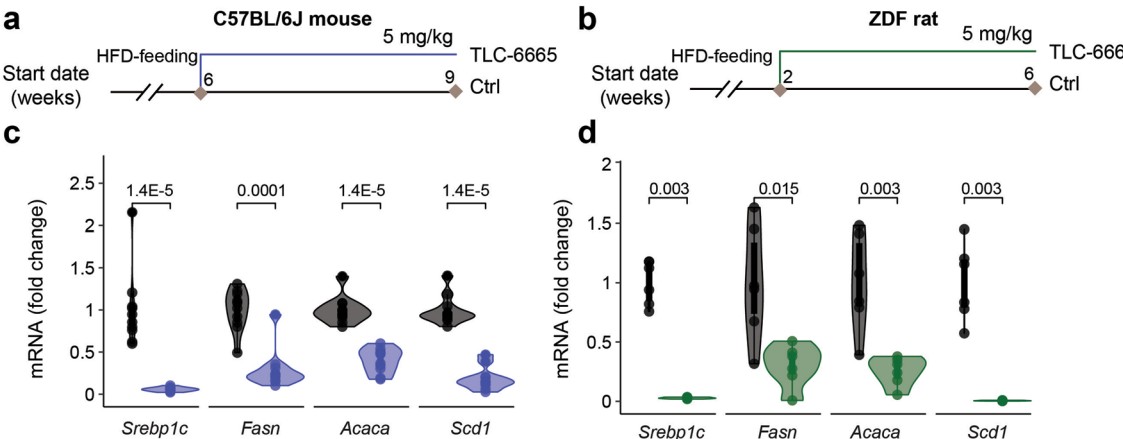

**Extended Data Fig. 4 | TLC-6665 reduces hepatic target gene expression**
*in vivo*. **(a-b)** Schematics illustrating evaluation of TLC-6665 in DIO mice **(a)**
and HFD-fed Zucker diabetic fatty (ZDF) rats **(b)**. **(c-d)** Liver target engagement
assessed by expression of LXR target genes in DIO mice **(c**, n = 10/group**)** and ZDF
rats **(d**, n = 6/group**)** after treatment with TLC-6665. Significance were calculated
by two-tailed Mann-Whitney U test and adjusted by BH adjustment.

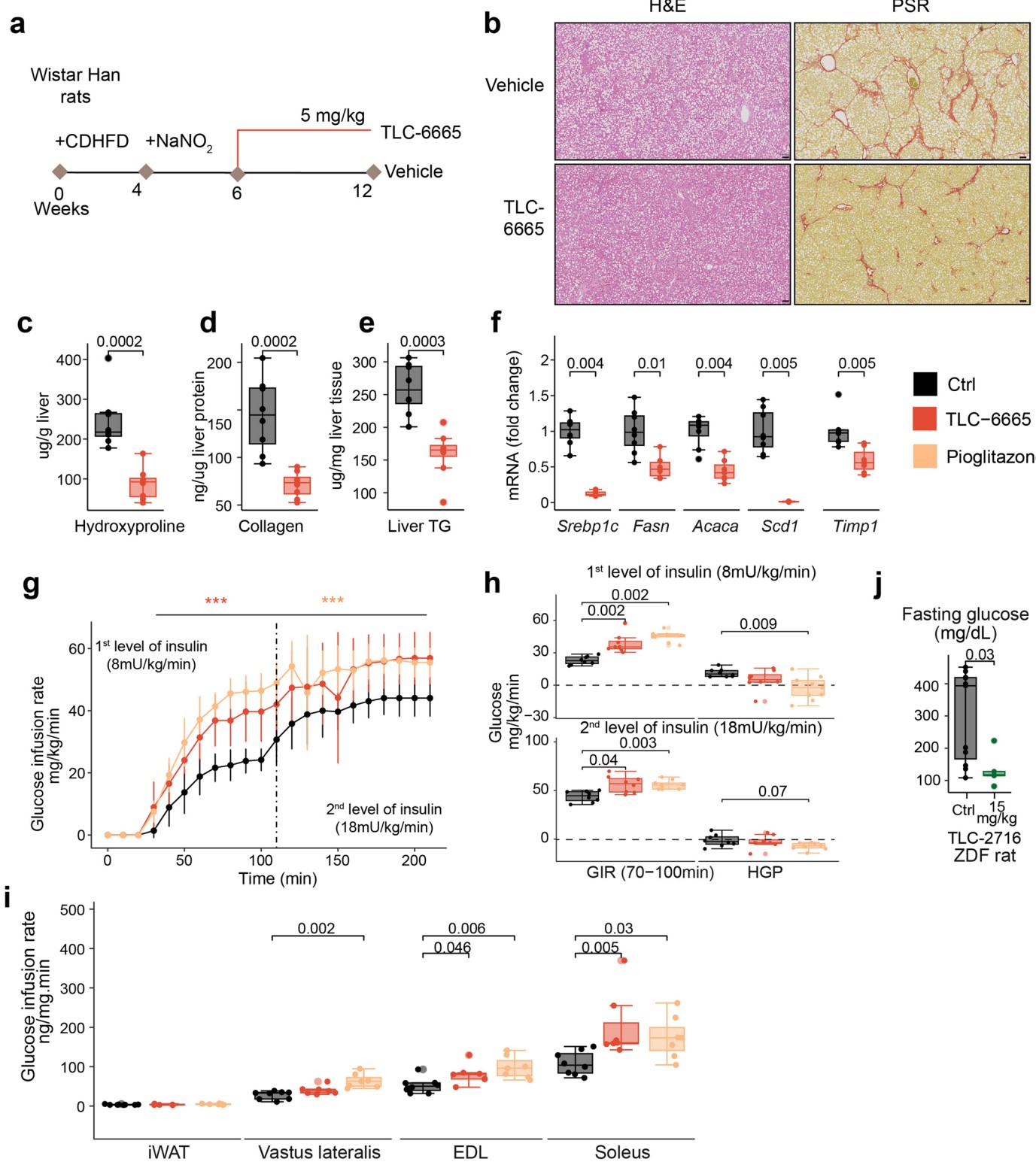

**Extended Data Fig. 5 | See next page for caption.**

**Extended Data Fig. 5 | LXR inverse agonists reduce the progression of liver fibrosis and improve insulin sensitivity.** (a) Schematic showing the experimental pipeline of MASH studies in Wistar Han rats. CDHFD: choline-deficient high fat diet. Rats were treated with vehicle or TLC-6665 (5 mg/kg) once daily by oral gavage for 6 weeks. (b) Representative images of hematoxylin and eosin (H&E) or Picrosirius red (PSR)-stained liver sections. Scale bar: 100μm. Images are representative of six biological replicates that showed similar results. (c-f) Measurements include liver hydroxyproline (c), liver collagen (d), liver TG content (e), and liver gene expression (f) from the MASH study in Wistar Han rats treated with vehicle or TLC-6665. n = 8/group (a-f). (g-h) Time course of glucose infusion rate (GIR, **g**, Data are shown as mean ± s.d.) and steady-state GIR and hepatic glucose production (HGP, **h**) during the two-step hyperinsulinemic-euglycemic clamp (the 1$^{st}$ [8 mU/kg/min] and 2$^{nd}$ [18 mU/kg/min] steps of insulin infusion) performed in DIO mice treated with vehicle, TLC-6665 (5 mg/kg) or pioglitazone (30 mg/kg) (i) Tissue glucose uptake measured at the end of the two-step hyperinsulinemic-euglycemic clamp performed in DIO mice treated with vehicle, TLC-6665, or pioglitazone for four weeks. EDL: extensor digitorum longus. n = 8/ctrl group, n = 8/TLC-6665 group, and n = 9/ Pioglitazone group (**g-i**). (**j**) Fasting plasma glucose levels after 5 weeks of treatment in HFD-fed ZDF rats dosed with vehicle (n = 11) or TLC-2716 (15 mg/kg, n = 5) once daily by oral gavage. *p < 0.05; **p < 0.01; ***p < 0.001. Two-tailed Mann-whitney U test and Benjamini-Hochberg (BH) adjustment (**f&h-i**). Two-tailed Mann-Whitney U test (**c-e**& **j**). Two-way ANOVA with Tukey Honest Significant Differences test (**g**). Box plots indicate the median (center line), interquartile range (IQR) (box bounds, 25th and 75th percentiles) and smallest and largest values within 1.5× IQR (whiskers) (**c-f**&**h-j**).

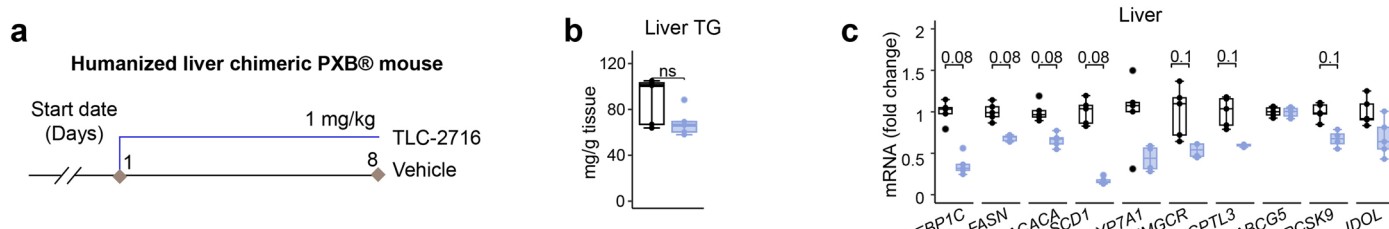

**Extended Data Fig. 6 | TLC-2716 maintains lipid homeostasis in a humanized liver murine model. (a)** Schematic for the evaluation of TLC-2716 in humanized liver chimeric PXB® mice. **(b, c)** Liver triglyceride (TG) content **(b)** and levels of liver transcripts involved in lipid synthesis and metabolism **(c)** in humanized liver chimeric PXB mice treated with TLC-2716 (1 mg/kg, once daily by oral gavage)

for 8 days. n = 5/group. P values were calculated by two-tailed Mann-Whitney U test and adjusted by Benjamini-Hochberg (BH) adjustment. Box plots indicate the median (center line), interquartile range (IQR) (box bounds, 25th and 75th percentiles) and smallest and largest values within 1.5× IQR (whiskers) **(b-c)**.

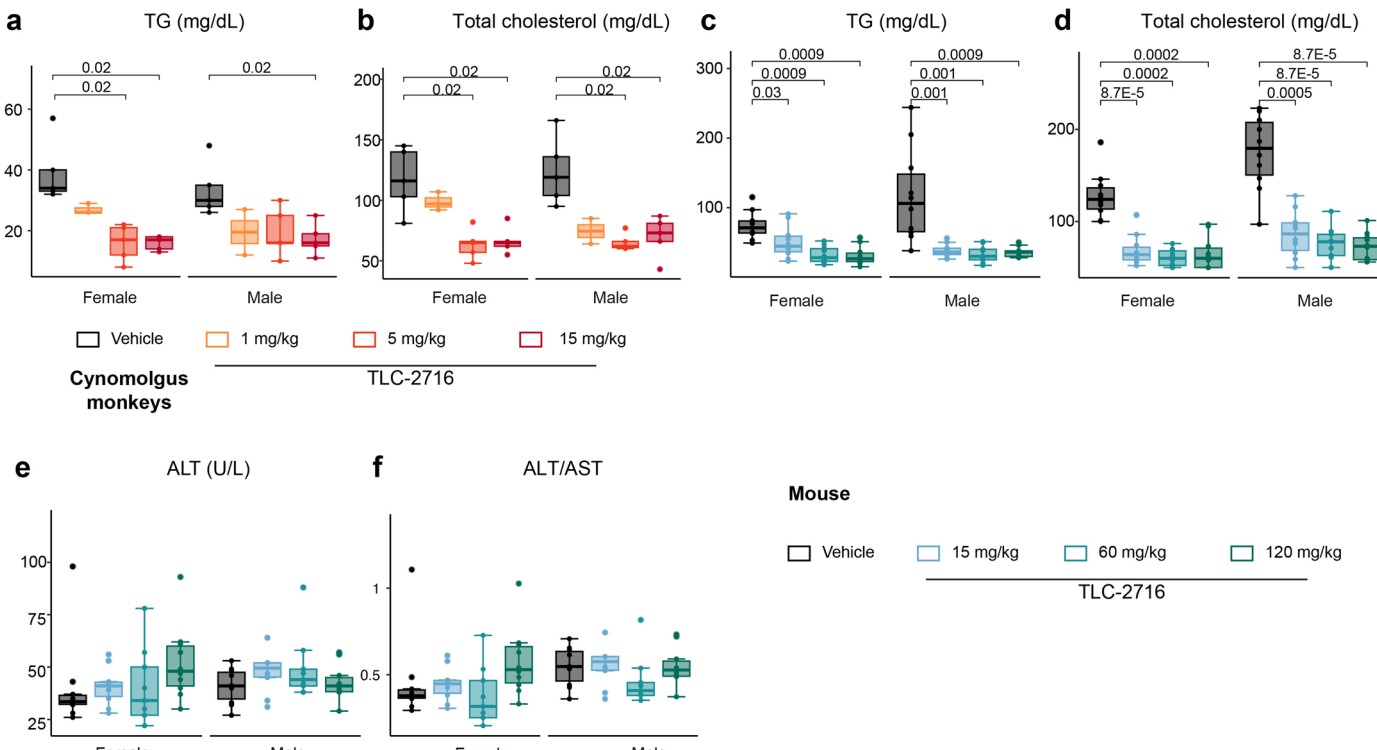

**Extended Data Fig. 7 | Effect of TLC-2716 on plasma lipids and hepatic safety in mice and in cynomolgus monkeys. (a-b)** Plasma TG **(a)** and TC **(b)** in a GLP toxicology study in cynomolgus monkeys treated with vehicle (n = 5/sex) or TLC-2716 1 mg/kg (n = 2 for male, n = 3 for female), 5 mg/kg (n = 5/sex), and 15 mg/kg (n = 5/sex) once daily by oral gavage for 28 days. **(c-f)** Plasma TG **(c)**, TC **(d)**, ALT **(e)**, and the ratio between ALT and AST **(f)** in CD-1 mice treated with 26 weeks of vehicle or TLC-2716 (15, 60, and 120 mg/kg) once daily by oral gavage. n = 10/group. P values were calculated by two-tailed Mann-Whitney U test and adjusted by BH adjustment. Box plots indicate the median (center line), interquartile range (IQR) (box bounds, 25th and 75th percentiles) and smallest and largest values within 1.5× IQR (whiskers) **(a-f)**.

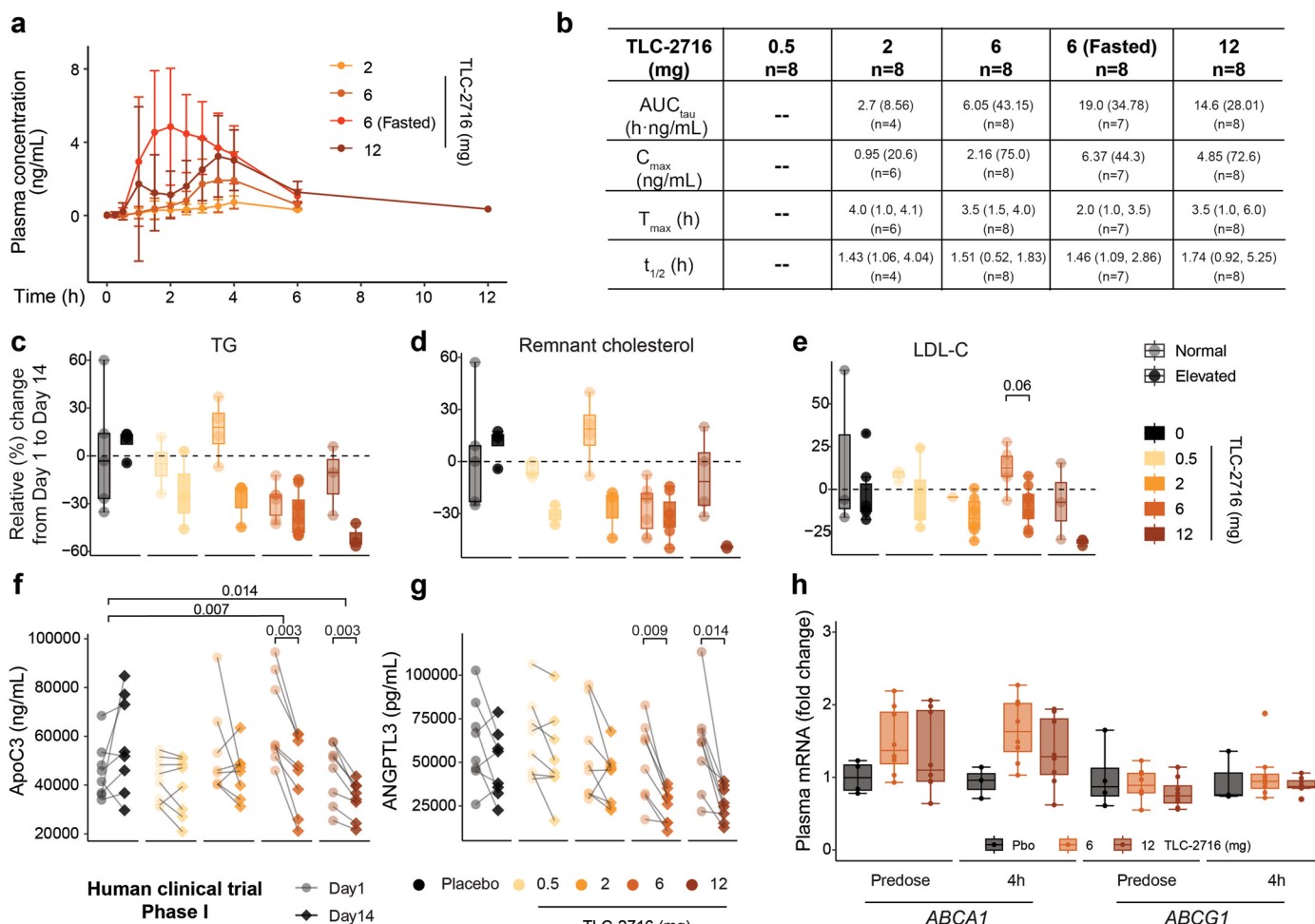

**Extended Data Fig. 8 | PK analysis of TLC-2716 and the effect of TLC-2716 on lipid metabolism in humans. (a-b)** Line plot showing the plasma exposure in humans treated with 2 mg (n = 6), 6 mg (n = 8/condition), or 12 mg (n = 8) TLC-2716 for 14 days **(a)** and the PK parameters for each dose group were indicated. **(b)**. TLC-2716 was administered in fed state unless indicated. PK parameters presented as mean (%CV) except $t_{1/2}$ and $T_{max}$, which are presented as median (min, max). **(c-e)** Box plots indicating the relative (%) change from Day 1 to Day 14 of TG **(c)**, remnant cholesterol **(d)**, and LDL-C **(e)** in individuals with normal or elevated baseline values: TG ($\geq$100 mg/dL), remnant cholesterol ($\geq$20 mg/dL), and LDL-C ($\geq$100 mg/dL), respectively. Significance was calculated by two-tailed Mann-Whitney U test and BH adjustment. Elevated TG: n = 4 (placebo), 3 (0.5 mg), 3 (2 mg), 8 (6 mg), 3 (12 mg). Normal TG: n = 5 (placebo), 4 (0.5 mg), 4 (2 mg), 6 (6 mg), 3 (12 mg). Elevated RC: n = 4 (placebo), 2 (0.5 mg), 3 (2 mg), 7 (6 mg), 2 (12 mg), Normal RC: n = 5 (placebo), 5 (0.5 mg), 4 (2 mg), 7 (6 mg), 4 (12 mg). Elevated LDL-C: n = 6 (placebo), 4 (0.5 mg), 6 (2 mg), 8 (6 mg), 3 (12 mg).

Normal LDL-C: n = 3 (placebo), 3 (0.5 mg), 1 (2 mg), 6 (6 mg), 3 (12 mg). **(f-g)** Dot plots showing the effect of 14-days feeding of TLC-2716 at 0.5 mg, 2 mg, 6 mg, or 12 mg on plasma levels of ApoC3 **(f**, n = 8/group)** and ANGPTL3 **(g**, n = 8/group)**. The significance of comparisons between Day 14 and Day 1 within the same dose group calculated by two-tailed Wilcoxon signed-rank test and BH adjustment, while the significance of changes of each parameter upon 14 days of treatment compared to placebo [(Day 14 − Day 1) / Day 1] calculated by two-tailed Mann-Whitney U test and BH adjustment. ANGPTL3 and ApoC3 were measured 12 h post dose in the postprandial state. **(h)** Boxplot showing the effect of TLC-2716 on plasma expression of genes involved in peripheral reverse cholesterol transport (*ABCA1* and *ABCG1*). Pbo: placebo. n = 4 (placebo-Predose), 3 (placebo-4h), 8/treatment group. Significance was calculated by two-tailed Mann-Whitney U test. Box plots indicate the median (center line), interquartile range (IQR) (box bounds, 25th and 75th percentiles) and smallest and largest values within 1.5× IQR (whiskers) **(c-e & h)**.

| Number (%) of Subjects with: | TLC-2716 | | | | | | Placebo |
|---|---|---|---|---|---|---|---|
| | 0.5 mg QD Fed (n = 8) | 2 mg QD Fed (n = 8) | 6 mg QD Fed (n = 8) | 12 mg QD Fed (n = 8) | 6 mg QD Fasted (n = 8) | Overall (n = 40) | (n = 10) |
| Any TEAE | 7 (87.5) | 3 (37.5) | 5 (62.5) | 3 (37.5) | 1 (12.5) | 19 (47.5) | 6 (60.0) |
| Diarrhea | 2 (25) | 0 (0) | 1 (12.5) | 2 (25) | 0 (0) | 5 (12.5) | 1 (10) |
| Headache | 1 (12.5) | 0 (0) | 1 (12.5) | 2 (25) | 0 (0) | 4 (10) | 2 (20) |
| Abdominal pain | 1 (12.5) | 0 (0) | 1 (12.5) | 1 (12.5) | 1 (12.5) | 4 (10) | 0 (0) |
| Back pain | 0 (0) | 0 (0) | 2 (25) | 0 (0) | 0 (0) | 2 (5) | 1 (10) |
| Pruritus | 4 (50) | 0 (0) | 0 (0) | 0 (0) | 0 (0) | 4 (10) | 0 (0) |
| Grade ≥ 2 TEAE | 0 (0) | 1 (12.5) | 0 (0) | 0 (0) | 0 (0) | 1 (12.5) | 0 (0) |
| Treatment-related TEAE | 0 (0) | 0 (0) | 1 (12.5) | 2 (25) | 0 (0) | 3 (7.5) | 1 (10) |
| TEAE leading to study drug discontinuation | 0 (0) | 0 (0) | 0 (0) | 0 (0) | 0 (0) | 0 (0) | 0 (0) |
| Serious TEAE | 0 (0) | 0 (0) | 0 (0) | 0 (0) | 0 (0) | 0 (0) | 0 (0) |
| Death | 0 (0) | 0 (0) | 0 (0) | 0 (0) | 0 (0) | 0 (0) | 0 (0) |

**Extended Data Fig. 9 | Safety assessment of TLC-2716 in humans.** Overall summary of treatment-emergent adverse events (TEAEs) in Phase 1 study of TLC-2716 in 50 healthy subjects involved in the MAD cohort.

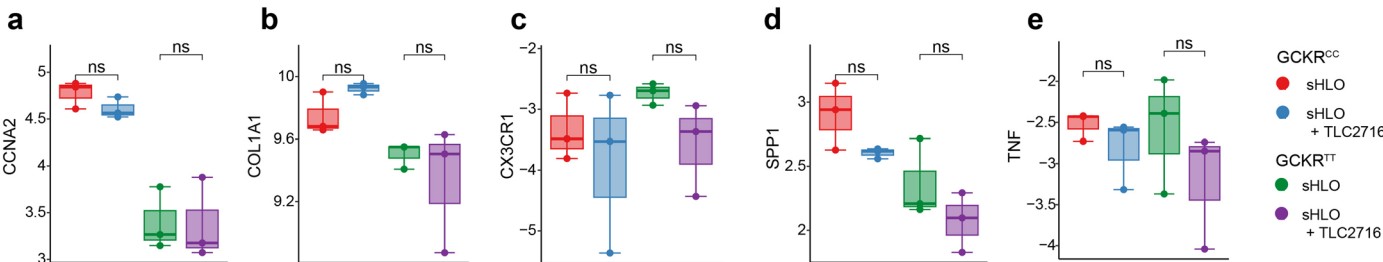

**Extended Data Fig. 10 | TLC-2716 does not exhibit pro-inflammatory or pro-fibrotic effects in steatotic human liver organoids (sHLOs).**
**(a-e)** Normalized gene expression of pro-inflammatory or pro-fibrotic genes in two subtypes of sHLOs, defined by the presence of variants in GCKR (CC vs TT). n = 3 samples per group. P values were calculated by two-tailed Student's t-test and BH-adjusted P values were indicated as follows: ns (not significant) BH-adjusted P value > 0.05. Box plots indicating the median (center line), interquartile range (IQR) (box bounds, 25th and 75th percentiles) and smallest and largest values within 1.5× IQR (whiskers) **(a-e)**.

# Reporting Summary

## Statistics

For all statistical analyses, confirm that the following items are present in the figure legend, table legend, main text, or Methods section.

| n/a | Confirmed | |
|---|---|---|
| ☐ | ☒ | The exact sample size (*n*) for each experimental group/condition, given as a discrete number and unit of measurement |
| ☐ | ☒ | A statement on whether measurements were taken from distinct samples or whether the same sample was measured repeatedly |
| ☐ | ☒ | The statistical test(s) used AND whether they are one- or two-sided *Only common tests should be described solely by name; describe more complex techniques in the Methods section.* |
| ☐ | ☒ | A description of all covariates tested |
| ☐ | ☒ | A description of any assumptions or corrections, such as tests of normality and adjustment for multiple comparisons |
| ☐ | ☒ | A full description of the statistical parameters including central tendency (e.g. means) or other basic estimates (e.g. regression coefficient) AND variation (e.g. standard deviation) or associated estimates of uncertainty (e.g. confidence intervals) |
| ☐ | ☒ | For null hypothesis testing, the test statistic (e.g. $F$, $t$, $r$) with confidence intervals, effect sizes, degrees of freedom and $P$ value noted *Give P values as exact values whenever suitable.* |
| ☒ | ☐ | For Bayesian analysis, information on the choice of priors and Markov chain Monte Carlo settings |
| ☐ | ☒ | For hierarchical and complex designs, identification of the appropriate level for tests and full reporting of outcomes |
| ☐ | ☒ | Estimates of effect sizes (e.g. Cohen's *d*, Pearson's *r*), indicating how they were calculated |

*Our web collection on statistics for biologists contains articles on many of the points above.*

## Software and code

Policy information about availability of computer code

| Data collection | The following software were used for data collection: VictorX4 multiplate reader (PerkinElmer Life Science) ; BMG-luminometer; quantitative real-time PCR (qRT-PCR) ; RNA was sequenced by BGI with the DNBSEQ platform. |
|---|---|
| Data analysis | The following software were used for data analysis: r studio (v.4.4.2); FastQC (version 0.11.9); STAR (version 2.73a); limma R package (version 3.50.3); clusterProfiler R package (version 4.2.2); TwoSampleMR R package (v0.6.2); REGENIE; Plink (genetics.binaRies R package, version 0.1.1 ); msigdbr R package (version 7.5.1) |

For manuscripts utilizing custom algorithms or software that are central to the research but not yet described in published literature, software must be made available to editors and reviewers. We strongly encourage code deposition in a community repository (e.g. GitHub). See the Nature Portfolio guidelines for submitting code & software for further information.

# Data

Policy information about availability of data

All manuscripts must include a data availability statement. This statement should provide the following information, where applicable:
- Accession codes, unique identifiers, or web links for publicly available datasets
- A description of any restrictions on data availability
- For clinical datasets or third party data, please ensure that the statement adheres to our policy

The study protocol and statistical plan are available within the article and supplemental information. The raw data for preclinical experiments and the summary result for the clinical trial phase 1 data is in the source data, while the individual data from the phase 1 clinical trial are available upon reasonable request from academic or qualified clinical researchers affiliated with recognized institutions, strictly for the purpose of conducting non-commercial, ethically approvable research aligned with the original scope of the trial. Applicants are required to submit a detailed research proposal, curriculum vitae and a declaration of non-conflict of interest. Requests must clearly describe the research objectives and methodology and must be reviewed and approved by corresponding authors. All approved requestors will be required to sign a data access agreement that restricts data use solely to the approved research project and prohibits any further distribution. The HLO RNAseq data are available under the GEO numbers GSE299888.

# Research involving human participants, their data, or biological material

Policy information about studies with human participants or human data. See also policy information about sex, gender (identity/presentation), and sexual orientation and race, ethnicity and racism.

| | |
|---|---|
| Reporting on sex and gender | This study only considered sex in the data analyses. For UK biobank data, this information was provided by the UK Biobank. For clinical trial, sex was self-reporting. |
| Reporting on race, ethnicity, or other socially relevant groupings | For the UK biobank data, only European-ancestry participants were considered. For clinical trial, we did not ask participants questions regarding race, ethnicity or other social relevant groupings |
| Population characteristics | For the UK biobank data, the population characteristics were provided by the UK Biobank. For 50 healthy subjects in the phase 1 clinical trial, 78% subjects were male. The mean age is 29.4 ± 8.9 years and mean BMI is 24.5 ± 4.1 kg/m2. This information is also included in the Table 1. |
| Recruitment | For the UK biobank data, Individuals were recruited by the UK Biobank. For clinical trial, individuals were evaluated by the investigator or staff at the study site. Recruitment was limited to locations or regions where the study being conducted. Written informed consent was obtained before patient enrollment. Eligible study subjects were healthy, non-smoking males and females between 18 and 55 years of age, and with BMI from 19 to 35 kg/m2, inclusive at screening. All subjects had estimated glomerular filtration rate ≥ 80 mL/min, normal liver biochemistry (total bilirubin 1.0- to 1.5-fold the upper limit of normal was permitted in subjects with Gilbert's syndrome), and 12-lead electrocardiograms (ECG) and screening laboratory evaluations (e.g., hematology, chemistry, and urinalysis) that were normal or considered to have no clinical significance by the investigator. In the MAD cohorts, an attempt was made to enroll subjects with TG ≥ 150 mg/dL and/or LDL-C ≥ 130 mg/dL to enable preliminary assessment of the lipid-lowering benefits of TLC-2716. Key exclusion criteria included pregnant or lactating females, TG ≥ 500 mg/dL, LDL-C ≥ 190 mg/dL, the presence of serious active medical or psychiatric illness, excessive alcohol consumption (defined as greater than 21 units/week for men and 14 units/week for women), substance abuse, or recent receipt of an investigational compound. Subjects who had taken any prescription or over-the-counter medications, including herbal products, within 28 days prior to the start of study drug dosing, except vitamins, acetaminophen, ibuprofen, and/or hormonal contraceptives, were excluded. A complete list of inclusion and exclusion criteria is available from the authors and study protocol. |
| Ethics oversight | For the UK biobank data, we have been allowed to use the UK Biobank Resource under Application Number 48020. For the clinical trial, the study was approved by the Northern B Health and Disability Ethics Committee (2022 FULL 12858) and conducted at New Zealand (Auckland Clinical Research) from 2022-07-27 to 2023-06-18 |

Note that full information on the approval of the study protocol must also be provided in the manuscript.

# Field-specific reporting

Please select the one below that is the best fit for your research. If you are not sure, read the appropriate sections before making your selection.

☒ Life sciences      ☐ Behavioural & social sciences      ☐ Ecological, evolutionary & environmental sciences

For a reference copy of the document with all sections, see nature.com/documents/nr-reporting-summary-flat.pdf

# Life sciences study design

All studies must disclose on these points even when the disclosure is negative.

| | |
|---|---|
| Sample size | No statistical method was used to determine sample size. Sample size was based on our previous publications: JHEP reports : innovation in |

| | |
|---|---|
| Sample size | hepatology, 5(9), 100815.; Journal of hepatology, 82(2), 174–188 |
| Data exclusions | For the plasma biochemistry data, they might have variations and outliers should be excluded. Outliers defined by Quantile-based method were excluded. The exclusion criteria were pre-established. |
| Replication | For animal studies they were performed once, but included biological replicates.<br>For clinical trial, no replication was conducted.<br>All other experiments were repeated at least two times and similar results are acquired. |
| Randomization | In animal studies, all animals were randomly allocated into each condition<br>The first-in-human, randomized, placebo-controlled Phase 1 study (NCT05483998), conducted at a single site in New Zealand (Auckland Clinical Research), included single-ascending dose (SAD) and multiple-ascending dose (MAD) cohorts. In the SAD cohorts, healthy subjects were treated with single oral doses of TLC-2716 (0.5, 2, 6, 12, and 20 mg) or placebo, and in the MAD cohorts, subjects received once-daily oral doses of TLC-2716 (0.5, 2, 6, and 12 mg) or placebo for 14 days. For each cohort, 8 subjects were randomized to receive TLC-2716 and 2 subjects to receive placebo. |
| Blinding | For the animal experiments, the investigators were not blinded to group allocation during data collection and analysis. Because investigators need to provide treatment to animal models, but individuals generated the RNAseq data do not know the group information.<br>For the clinical trial, the outcome assessors and the research assistant handling the data were blinded. Computational analysis was not performed blinded. |

# Reporting for specific materials, systems and methods

We require information from authors about some types of materials, experimental systems and methods used in many studies. Here, indicate whether each material, system or method listed is relevant to your study. If you are not sure if a list item applies to your research, read the appropriate section before selecting a response.

## Materials & experimental systems

| n/a | Involved in the study |
|---|---|
| ☒ | ☐ Antibodies |
| ☒ | ☐ Eukaryotic cell lines |
| ☒ | ☐ Palaeontology and archaeology |
| ☐ | ☒ Animals and other organisms |
| ☐ | ☒ Clinical data |
| ☒ | ☐ Dual use research of concern |
| ☒ | ☐ Plants |

## Methods

| n/a | Involved in the study |
|---|---|
| ☒ | ☐ ChIP-seq |
| ☒ | ☐ Flow cytometry |
| ☒ | ☐ MRI-based neuroimaging |

## Animals and other research organisms

Policy information about studies involving animals; ARRIVE guidelines recommended for reporting animal research, and Sex and Gender in Research

| | |
|---|---|
| Laboratory animals | 18-week old male C57BL/6 diet-induced obese (DIO) mice (14 weeks on high-fat diet, Research Diets, New Jersey, USA), purchased from Jackson Laboratories (Maine, USA); 6- to 7-week-old male obese (fa/fa) Zucker rats (ZUCKER-Leprfa), 6- to 7-week-old male Sprague Dawley rats, 6- to 8-week-old male Wistar rats bought from Charles River Laboratories; A 26-week Good Laboratory Practice (GLP) toxicology study were performed with CD-1 mice (Charles River Laboratories) and A 4-week Good Laboratory Practice (GLP) toxicology study was conducted in cynomolgus monkeys (Guangzhou Xiangguan Biotech Co., Ltd). Human liver chimeric PXB® mice were purchased from PheonexBio (Japan), and in-life procedures were performed at InterVivo Solution (Ontario, Canada). |
| Wild animals | Study did not involve wild animals |
| Reporting on sex | Male animals were used in this study for the functional test experiments to limit the number of animals. For the toxicity study, we used both male and female animals |
| Field-collected samples | Study did not involve field-collected samples. They were all studied in the lab. |
| Ethics oversight | the in-vivo studies were performed at Synovo GmbH (Tübingen, Garmany) in accordance with their bioethical guidelines, which are fully compliant to ethical regulations and internationally accepted principles for the care and use of laboratory animals;<br>the study performed at InterVivo Solution (Ontario, Canada) in accordance with their bioethical guidelines, which are fully compliant to ethical regulations and internationally accepted principles for the care and use of laboratory animals. The study performed at Physiogenex S.A.S. (Labège, France) in accordance with the ethical regulations, Guide for the Care and Use of Laboratory Animals (revised 1996 and 2011, 2010/63/EU) and French laws. All procedures involving animals were reviewed and approved by the Institutional Animal Care and Use Committee (IACUC) and conducted in accordance with international guidelines for the care and use of laboratory animals. |

Note that full information on the approval of the study protocol must also be provided in the manuscript.

# Clinical data

Policy information about clinical studies

All manuscripts should comply with the ICMJE guidelines for publication of clinical research and a completed CONSORT checklist must be included with all submissions.

| | |
|---|---|
| Clinical trial registration | ClinicalTrials.gov (NCT05483998) |
| Study protocol | The study protocol are available in the method section and supplemental information |
| Data collection | This study is registered at ClinicalTrials.gov (registration: NCT05483998) and was conducted at a single site in New Zealand (Auckland Clinical Research) from 2022-07-27 to 2023-06-18, in accordance with relevant local regulatory policies. Objectively measured outcomes were collected at baseline and 14 days, including predose, 0.5, 1, 1.5, 2, 2.5, 3, 3.5, 4, 6, and 12 hours after treatment at the intervention sites by blinded assessors |
| Outcomes | Primary Outcome Measures:<br>(1) Number of subjects with treatment-emergent adverse events (TEAEs) in single ascending dose (SAD) compared to placebo.<br>(2) Number of subjects with clinically significant change from Baseline in vital signs in SAD.<br>(3) Number of subjects with laboratory abnormalities in SAD.<br>(4) Number of subjects with electrocardiogram (ECG) abnormalities in SAD.<br>(5) Number of subjects with TEAEs in multiple ascending dose (MAD) compared to placebo.<br>(6) Number of subjects with clinically significant change from Baseline in vital signs in MAD.<br>(7) Number of subjects with laboratory abnormalities in MAD.<br>(8) Number of subjects with ECG abnormalities in MAD.<br>Secondary Outcome Measures:<br>(1) Plasma concentration of each dose of study drug to determine AUClast in SAD.<br>(2) Plasma concentration of each dose of study drug to determine AUCinf in SAD.<br>(3) Plasma concentration of each dose of study drug to determine %AUCexp in SAD.<br>(4) Plasma concentration of each dose of study drug to determine CL/F in SAD.<br>(5) Plasma concentration of each dose of study drug to determine Cmax in SAD.<br>(6) Plasma concentration of each dose of study drug to determine Tmax in SAD.<br>(7) Plasma concentration of each dose of study drug to determine Clast in SAD.<br>(8) Plasma concentration of each dose of study drug to determine Tlast in SAD.<br>(9) Plasma concentration of each dose of study drug to determine t1/2 in SAD.<br>(10) Plasma concentration of each dose of study drug to determine λz in SAD.<br>(11) Plasma concentration of each dose of study drug to determine AUClast in MAD.<br>(12) Plasma concentration of each dose of study drug to determine AUCtau in MAD.<br>(13) Plasma concentration of each dose of study drug to determine Ctau in MAD.<br>(14) Plasma concentration of each dose of study drug to determine CLss/F in MAD.<br>(15) Plasma concentration of each dose of study drug to determine Cmax in MAD.<br>(16) Plasma concentration of each dose of study drug to determine Tmax in MAD.<br>(17) Plasma concentration of each dose of study drug to determine Clast in MAD.<br>(18) Plasma concentration of each dose of study drug to determine Tlast in MAD.<br>(19) Plasma concentration of each dose of study drug to determine t1/2 in MAD.<br>(20) Plasma concentration of each dose of study drug to determine λz in MAD. |

# Plants

| | |
|---|---|
| Seed stocks | *Report on the source of all seed stocks or other plant material used. If applicable, state the seed stock centre and catalogue number. If plant specimens were collected from the field, describe the collection location, date and sampling procedures.* |
| Novel plant genotypes | *Describe the methods by which all novel plant genotypes were produced. This includes those generated by transgenic approaches, gene editing, chemical/radiation-based mutagenesis and hybridization. For transgenic lines, describe the transformation method, the number of independent lines analyzed and the generation upon which experiments were performed. For gene-edited lines, describe the editor used, the endogenous sequence targeted for editing, the targeting guide RNA sequence (if applicable) and how the editor was applied.* |
| Authentication | *Describe any authentication procedures for each seed stock used or novel genotype generated. Describe any experiments used to assess the effect of a mutation and, where applicable, how potential secondary effects (e.g. second site T-DNA insertions, mosiacism, off-target gene editing) were examined.* |

