## [Peer Review File · Nature Medicine]

An oral, liver-restricted LXR inverse agonist for dyslipidemia: preclinical development and phase 1 trial

Corresponding Author: Professor Johan Auwerx

Version 1:

Reviewer comments:

Reviewer #1

(Remarks to the Author)

Review for "Liver-specific LXR inverse agonist restores lipid homeostasis in animal models and humans"

Li et al. demonstrated that inhibiting liver X receptor (LXR) improves lipid metabolism and insulin sensitivity. They developed TLC-2716, a gut- and liver-restricted LXR inverse agonist, which reduced lipid accumulation in human liver organoids and safely lowered plasma triglycerides and remnant cholesterol in a Phase 1 trial. These results highlighted TLC-2716 as a promising therapy for dyslipidemia and residual ASCVD risk. The study was well-designed with detailed evidence. However, there are some additional major issues that need to be addressed, including statistical models use, lack of multiple-comparison, detailed report for phase I clinical trial results, and more evidence from post GWAS analysis.

Major:

1. Authors used several GWAS cohorts including UKB, FinnGen, and MVP to validate the finding that "genetic variants within LXR α , but not LXR, are associated with lipid biomarkers". It is unclear how these genetic variants were located into genes in the study. Due to the structure and complexity of the genome, GWAS signals are often located in non-coding regions and influenced by LD structure, suggesting that the identified associations may reflect regulatory elements rather than direct effects on protein-coding sequences. It would be great that authors can perform some additional post-hoc GWAS analysis, including fine-mapping to better pinpoint the causal signals to the gene. TWAS could also help to identify relevant tissues for the association. Additionally, since the manuscript was aiming to establish a potential causal relationship between LXR α and metabolic disorders, it would be helpful to utilize Mendelian randomization method to try to assess causal relationship, using genetics as instrumental variables.
2. Figure 2: authors presented comparison within each type of mice. (1) A clear illustration of what the black lines mean in b to e is needed. (2) Two-way ANOVA were used for comparison among 3 groups with different dose. It is unclear what time point two-way ANOVA is comparing. It seems like it is only comparing at the last time point. A better approach is to compare the trend of these three trajectories. Also, what covariates are considered in the model? (3) When looking at the measurements among the 3 groups at time 0, some values were not close to each other (e.g., left panel of Figure 2b, middle panel of Figure 2e). From the study design, these mice should have been treated under the same condition across the three dose groups. Some explanations are needed for the variability at time 0. Since the baseline values are significantly different between groups, please also address the potential lack of randomization between the groups. If available, it would be helpful to include baseline measurements taken before the feeding period as a reference.
3. Section "LXR inverse agonists mitigate dyslipidemia-related metabolic disorders" and figure 3: author used two-tailed Student's t-test to compare between two groups within each type of mouse. There are a few potential issues: (1) sample size for each group. If the sample size is small, it might result in inflated type I error. (2) Are there any potential confounding variables would influence the results? Student's t-test cannot take into consideration of potential confounding factors, like height, weight etc.
4. A phase 1 clinical trial was illustrated in the study to prove "TLC-2716 was safe and improved multiple atherogenic lipid parameters in healthy subjects": generally speaking, more detailed information should be reported for this phase 1 clinical trial. More specifically: (1) authors mentioned 50 healthy individuals were randomized for this clinical trial into several groups, include "either placebo (2 subjects per dose) or TLC-2716 0.5, 2, 6, or 12 mg (8 subjects per dose)". It would be important to have a baseline characteristic comparison to make sure there are no significant differences for covariates of interest. Currently, only a study level summary is provided. (2) There were some mild adverse events reported, it would be great to conduct a sensitivity analysis by excluding those adverse events to examine the robustness of results.

5. Multiple comparison: there were many places that involved potential multiple comparison. Although authors indicated significance by use different notations to indicate different levels of significant, it is important to perform multiple comparison adjustment to avoid inflated type I error. Additionally, it would be great to report p-value or/and adjusted p-value as well.

Minor:

1. Typo "sush" in "...sush as Fasn, Scd1, and Acaca,...".

2. Phase 1 clinical trial section, a citation might be needed for "TLC-2716 was first tested in a single-ascending dose (SAD) study in healthy subjects in which it was shown to be safe and well tolerated in single doses up to 20 mg."

3. GWAS study signals are identified using widely use p-value threshold of 5e-08 to consider for multiple comparisons. In the method section, authors used 1e-08 as the threshold. It would be great that author can justify a more stringent threshold was used here. A more stringent threshold might lead to missing signals. Please clarify the choice of this arbitrary threshold.

In all, this study presented compelling evidence that the gut- and liver-restricted LXR inverse agonist TLC-2716 improves lipid metabolism and showed promise as a therapeutic strategy for dyslipidemia and residual ASCVD risk. While the study was well-designed and provided valuable preclinical and early clinical insights, several major issues limited the strength of the conclusions, including the need for additional post-GWAS analyses (fine-mapping, TWAS, Mendelian randomization), clarification and refinement of statistical models in animal studies, baseline and covariate adjustments in the clinical trial, detailed description of phase I clinical trial results, and proper multiple-comparison corrections. Addressing these concerns, along with minor revisions for clarity and consistency, would greatly strengthen the manuscript and its translational impact.

Reviewer #2

(Remarks to the Author)

The uniqueness of this study is based on the irony of the efficacy (in humans) of compounds that target LXR for suppression of LXR target genes (LXR inverse agonists), rather than LXR agonists that have been a major focus of therapeutic development for decades. Beginning in 2000, LXR was shown to increase ABCA1 expression and reverse cholesterol transport (RXT) (PMID: 10858438; PMID: 11790770) and in 2002 LXR agonists were first shown to have efficacy in reduction of atherosclerosis (PMID: 12032330) leading to a substantial effort to develop such synthetic agonists to treat heart disease. However, about the same time, synthetic LXR agonists were also shown to increase hepatic lipogenesis leading to fatty liver and elevated plasma levels of TGs via induction of key genes such as SREBP1, FAS, and SCD1 (PMID: 11090131). This side effect limited the potential clinical utility of LXR agonists. This led to years of attempts to develop LXR agonists that could increase RCT while having no effect on fatty liver and TGs. LXR agonists with favorable preclinical profiles were developed but displayed challenging data in humans indicating that although LXR agonism drove an increase in HDL-C, it also was associated with an increase in TGs and unexpectedly, an increase in LDL-C as well (PMID: 27508871).

In 2013, with the knowledge that LXR activation was very effective in induction of lipogenesis, the Burris group focused on developing the first LXR inverse agonists that would selectively recruit corepressors to LXR resulting in suppression of LXR target genes leading to reduction in hepatic lipogenesis. Understanding that these compounds would likely also inhibit RCT they developed tool compounds that displayed liver selectivity, and these compounds were efficacious in animal models of fatty liver (PMID: 27508871), MASH (PMID: 25830098), and ASH (PMID: 31696159; PMID: 33770118) without affecting RCT in the periphery. These first-generation compounds did not have oral bioavailability, but demonstrated the feasibility of a liver targeted LXR inverse agonist. These first-generation compounds also had beneficial effects on insulin sensitivity, plasma TGs, and LDL-C levels (PMID: 25830098; PMID: 35417135).

In this manuscript, Li et al describe the profile and development of LXR inverse agonist for treatment of dyslipidemia and MASH and results of clinical data demonstrating their effectiveness in humans. The manuscript describes the preclinical development of 2 LXR inverse agonists and further clinical data (phase 1) with one selected LXR inverse agonist with a favorable PK/PD and toxicology profile. Preclinical data includes an array of biochemical, cell based, and rodent models. The profile is impressive and the lead compound that proceeded to clinical trials displayed outstanding safety data, selective liver exposure, and outstanding efficacy in lowering plasma TG and LDL-C. Beyond the expected suppression of genes that encode proteins essential for lipogenesis, a very interesting effect of this class of compound is its pleiotropic effect on expression of a range of genes whose products are targets for dyslipidemia themselves including HMGCR, PCSK9, IDOL, ANGPTL3, and APOC3. There is significant promise for such a compound for treatment of dyslipidemia, fatty liver and MASH in humans and the results are quite exciting. Below are several areas that should be addressed:

1. In the introduction, the authors use the term "downregulation" of LXR activity, and this term is not a sufficient method to describe an inverse agonist. Downregulation suggest that the effect of a compound is to reduce the efficacy of activation of a receptor. This is not what is occurring since the inverse agonist is causing an active repression of target genes and not merely downregulating the ability of genes to be activated.
2. No mention of previous development of LXR inverse agonists or liver selective LXR inverse agonists is mentioned in the introduction. As mentioned above, the preclinical proof-of-principle of liver selective LXR inverse agonists has been conducted in models of MASH and dyslipidemia previously. This should be referenced in the introduction.
3. On line 130, the authors describe the TLC compounds as "first-in-class" LXR inverse agonists. These are not the first LXR inverse agonists discovered and tested in preclinical models, but one is the first in humans. This should be clarified.
4. I'm unable to find the specifics of how many of the biochemical and cell-based assays were performed. The binding assay and transfection assays have no methods. Were the binding assays radioligand binding displacement assays? If so, what was the radioligand utilized to displace? Ki values would be more appropriate than EC50 values. It would also be useful to

have a comparator compound such as SR9238 to compare the activity of the TLC compounds to in these assays.

5. In the evaluation of the 2716 in the rodent models (beginning with line 142) was there plasma exposure data used to select the doses of 0.3 and 1 mg/kg/day. Given the relatively short half-life, was the compound ever examined with b.i.d. dosing?

6. On line 255, the authors describe the liver selective exposure. Have liver partition coefficients been calculated (K_p , liver) for each of these compounds?

7. In Figure 3i, it appears that there are two labels for the y axis- Fasting Glucose and mRNA

8. In extended figure 6a, it appears that there is a very large food effect on plasma drug levels (based on the 6 mg dose). Was this the case? If so, was this expected based on the previous PK studies?

Reviewer #4

(Remarks to the Author)

I read with interest this investigation of LXR in lipid metabolism, insulin sensitivity and metabolic liver disease by Li and colleagues. They have presented a comprehensive series of studies that include genome wide association studies from multiple cohorts, metabolic mouse models of both dyslipidemia and liver disease and then studies leading to a phase 1 trial of a LXR inverse agonist. Essentially the findings consistently implicate LXR in lipid metabolism, favorable effects in mouse models and organoid studies on dyslipidemia, insulin sensitivity and MALSD and a phase 1 trial demonstrating the LXR inverse agonist being well tolerated, having beneficial effects on atherogenic lipid parameters and improving surrogate markers of insulin sensitivity. The selective inverse agonist designed to work within the liver and gut, not systemically in order to avoid potential adverse effects on reverse cholesterol transport. The strength of the manuscript lies in the comprehensive series of studies presented. The authors should consider a number of points.

1. There is a large use of surrogate readouts. A lot of gene regulation results, without accompanying functional data – the reverse cholesterol transport data is a good example, with only ABCA1 presented as if that is a perfect correlate of transport, which it is not – similarly, the surrogates of insulin sensitivity in the phase 1 trial. There are many more. This in my mind softens the strength of the paper.

2. The field of LXR agonists has proven challenging in drug development, largely due to broad pleiotropic effects, safety issues and modest metabolic efficacy. The authors imply that a potential adverse effect on reverse cholesterol transport is the problem – it is ONE problem. A more balanced view of the previous challenges in this field and how they propose to distinguish the findings from those programs would be important to do.

3. The authors should tone down comments as “safety” in the phase 1 trial. They would be better to talk to tolerability and the presence or lack of adverse events.

4. The individuals selected for inclusion in the phase 1 trial had modest dyslipidemia at best. Given the range of triglyceride levels at baseline, none of these individuals would be candidates for this drug.

5. The findings are modest. The triglyceride lowering is far below what is being shown with inhibitors of ANGPTL3 and apoC3. The apoB reduction is far below what would be needed to progress this compound in development. The Mendelian randomization data is clear in demonstrating the relationship between apoB lowering and CV benefit in this space. I'm not convinced that there is enough here to be too optimistic that this agent will be a useful therapeutic in the prevention clinic. Much more work is required.

Version 2:

Reviewer comments:

Reviewer #4

(Remarks to the Author)

The authors are thanked for their responses. I remain concerned that the majority of findings are indirect at best and the phase 1 findings are underwhelming.

Comment to all reviewers:

Based on Reviewer #1's suggestion, we have now applied the Mann–Whitney U test to replace the student's t-test to determine the statistical significance for the small sample size datasets, such as the results for *in vivo* experiments. Benjamini–Hochberg (BH) adjustment was also performed in all relevant analyses to reduce the risk of type I error. While the p values have now slightly changed for the *in vivo* experiments, but the main identified treatment effect and conclusions remain unchanged. All figures and results have been updated with these changes. The responses are colored in blue. The modified text in the revised manuscript is highlighted in yellow. The page number and line number in this point-to-point rebuttal refer to the clean and untracked version of the revised manuscript, and the Extended Data Figures and Figure Legends file.

Reviewer #1 (Remarks to the Author):

Li et al. demonstrated that inhibiting liver X receptor (LXR) improves lipid metabolism and insulin sensitivity. They developed TLC-2716, a gut- and liver-restricted LXR inverse agonist, which reduced lipid accumulation in human liver organoids and safely lowered plasma triglycerides and remnant cholesterol in a Phase 1 trial. These results highlighted TLC-2716 as a promising therapy for dyslipidemia and residual ASCVD risk. The study was well-designed with detailed evidence. However, there are some additional major issues that need to be addressed, including statistical models use, lack of multiple-comparison, detailed report for phase I clinical trial results, and more evidence from post GWAS analysis.

Major:

1. Authors used several GWAS cohorts including UKB, FinnGen, and MVP to validate the finding that “genetic variants within $LXR\alpha$, but not LXR, are associated with lipid biomarkers”. It is unclear how these genetic variants were located into genes in the study. Due to the structure and complexity of the genome, GWAS signals are often located in non-coding regions and influenced by LD structure, suggesting that the identified associations may reflect regulatory elements rather than direct effects on protein-coding sequences. It would be great that authors can perform some additional post-hoc GWAS analysis, including fine-mapping to better pinpoint the causal signals to the gene.

Thank you for your suggestions. We have now extracted the fine-mapping result of lipid-related traits based on UKBB database¹ and found one causal signal of $LXR\alpha$ that associated with apolipoprotein A and HDL cholesterol (**Extended data Figure 2a**). We also performed a burden test for $LXR\alpha$ based on the UKBB whole exome sequence (WES) and found that the presence of the group of genetic variants (loss of function [LOF], minor allele frequency < 0.001) is linked to higher HDL cholesterol and lower triglycerides level. (**Extended data Figure 2b**). Consistently, the burden test based on the UKBB whole genome sequencing also indicated that LOF genetic variants within LXR has a large effect on the HDL cholesterol, triglycerides, ApoA1, ALT, and AST². Taken together, the genetic variants within the $LXR\alpha$ have a potential causal effect on lipid metabolism. Please find the **Extended data Figure 2** in revised Extended Data Figures and Figure Legends file, page 2.

TWAS could also help to identify relevant tissues for the association.

We appreciate this reviewers' concerns, but unfortunately, we cannot perform TWAS since there is no liver gene expression data in the UKBB. But we performed gene set enrichment analysis to explore the co-expressed gene sets of *LXRα* across tissues in both sexes using the human GTEx database. The analysis revealed that *LXRα* expression in adipose tissue, coronary artery, spleen, colon, liver, ileum, and whole blood (highlighted in red) was positively correlated with genes involved in amino acid, fatty acid, triglyceride, and cholesterol metabolism (Figure R1). These findings suggest that the gene expression of *LXRα* in these tissues are particularly relevant for lipid metabolism.

Figure R1: Geneset enrichment analysis highlights the co-expressed genesets of *LXRα* in both sexes across tissues in human GTEx database. NES: Normalized enrichment score; Significance: *q value <0.05; **q value <0.01; ***q value <0.001.

Additionally, since the manuscript was aiming to establish a potential causal relationship between *LXRα* and metabolic disorders, it would be helpful to utilize Mendelian randomization method to try to assess causal relationship, using genetics as instrumental variables.

In this study, we demonstrated a causal role of *LXRα* and *LXR* signaling in lipid metabolism. Following your suggestion, we first extracted significant *cis*-eQTLs of *LXRα* (NR1H3) and *LXRβ* (NR1H2), in liver from the GTEx (version 8)⁷, and we performed Linkage disequilibrium (LD) clumping to identify the independent SNPs as instrumental variables. However, due to the limited statistical power (208 individuals), only one independent SNP was found within *LXRα* using the GTEx liver *cis*-eQTLs, which is less reliable for MR analysis. But to our knowledge, no better liver *cis*-eQTLs resource is publicly available. To overcome this limitation, we extracted the *cis*-eQTLs of *LXRα* in whole blood (another potentially relevant tissue) from the eQTLGen database (31,684 individuals)⁸. Eight independent SNPs were identified using the eQTLGen blood *cis*-eQTLs after LD clumping. The number of

independent SNPs and that of participants in the two databases both suggest that the eQTLGen blood *cis*-eQTLs are more reliable and has more statistical power for MR analysis. Therefore, we applied significant *cis*-eQTLs of LXR in blood as exposures. GWAS summary statistics for plasma lipid-related traits were derived from whole-exome sequencing (WES) data in the UK Biobank and served as outcomes.

As shown in **Figure 1f** (*LXR α*), the MR analysis revealed that the expression of *LXR α* , but not *LXR β* (no significant associations), in blood is associated with plasma lipid-related traits. Although we cannot obtain a reliable causal effect of hepatic *LXR α* expression on lipid metabolism and metabolic disease by MR, we can still confirm the essential role of *LXR α* and its target genes in lipid metabolism and metabolic liver disease based on **Figure. 1d-e** (page 38 in the revised manuscript file) & **Extended Data Figure 2c** (page 2 in the revised Extended Data Figures and Figure Legends file).

2. Figure 2: authors presented comparison within each type of mice. (1) A clear illustration of what the black lines mean in b to e is needed.

We have now indicated the meaning of black lines in the figure legend of **previous Extended Data Figure 2 (current Extended Data Figure 3)**. Please find the modified text in the revised Extended Data Figures and Figure legends file (see page 3, line 42-43).

(2) Two-way ANOVA were used for comparison among 3 groups with different dose. It is unclear what time point two-way ANOVA is comparing. It seems like it is only comparing at the last time point. A better approach is to compare the trend of these three trajectories. Also, what covariates are considered in the model?

We apologize for the confusion. The significance indicated in the figure represents the difference between the treatment group and the vehicle group, calculated based on the trend of the three trajectories using two-way ANOVA. The model applied was lipid-related parameter \sim treatment condition + days + interaction between treatment condition and days, followed by Tukey's Honest Significant Difference test. To improve clarity, we have repositioned the asterisks and included this formula in the figure legend of **previous Extended Data Figure 2 (current Extended Data Figure 3)**. Please find the modified figure legends in the revised Extended Data Figures and Figure Legends file (see page 3, line 43-46).

(3) When looking at the measurements among the 3 groups at time 0, some values were not close to each other (e.g., left panel of Figure 2b, middle panel of Figure 2e). From the study design, these mice should have been treated under the same condition across the three dose groups. Some explanations are needed for the variability at time 0. Since the baseline values are significantly different between groups, please also address the potential lack of randomization between the groups. If available, it would be helpful to include baseline measurements taken before the feeding period as a reference.

Thank you for pointing this out. Plasma parameters are quite variable. Since these measurements were performed only at the end of the animal experiments, we could not randomize the mice based on the plasma parameters at the beginning of treatment.

We agree that baseline values should be taken into account in the analysis. To address this, we applied two approaches to improve our analysis: (1) we first used the area under the curve (AUC) to represent the overall changes in plasma lipid parameters (TG and TC) and inflammation markers (ALT and AST) following TLC-2716 or TLC-6665 treatment; and (2) we identified extreme outliers in the AUC values and applied a quantile-based method to remove them. We then replotted the boxplot figure in a **revised Figure 2e-f&h** (please find them in the revised manuscript, Page 39) and line plots in the **revised Extended data Figure 3** (please find them in the revised Extended Data Figures and Figure legends file, Page 3).

3. Section “LXR inverse agonists mitigate dyslipidemia-related metabolic disorders” and figure 3: author used two-tailed Student’s t-test to compare between two groups within each type of mouse. There are a few potential issues: (1) sample size for each group. If the sample size is small, it might result in inflated type I error.

We agree with your suggestion and have replaced the two-tailed Student’s t-test with the Mann–Whitney U test, which is more appropriate for datasets with small sample sizes in all relevant comparisons. Then, the Benjamini–Hochberg (BH) adjustment was performed in all relevant analyses to reduce the risk of type I error.

In addition, the sample size of each group was indicated in the figure legend of the revised **Extended Data Figure 5** (previous Figure 3). Please find the modified figure legends in the revised manuscript and Extended Data Figures and Figure legends file, page 5.

(2) Are there any potential confounding variables would influence the results? Student’s t-test cannot take into consideration of potential confounding factors, like height, weight etc. For the liver- and glucose infusion rate–related parameters, we adjusted for liver weight or body weight; we do not expect additional confounding variables that could affect the results since rodent experiments are performed on genetically identical animals in controlled environmental conditions.

4. A phase 1 clinical trial was illustrated in the study to prove “TLC-2716 was safe and improved multiple atherogenic lipid parameters in healthy subjects”: generally speaking, more detailed information should be reported for this phase 1 clinical trial. More specifically: (1) authors mentioned 50 healthy individuals were randomized for this clinical trial into several groups, include “either placebo (2 subjects per dose) or TLC-2716 0.5, 2, 6, or 12 mg (8 subjects per dose)”. It would be important to have a baseline characteristic comparison to make sure there are no significant differences for covariates of interest. Currently, only a study level summary is provided.

(1). Please find the baseline characteristics of all participants in **Figure 4b-c** (page 41 in the revised manuscript) and **Table 1** (page 26-27 in the revised manuscript). We also performed comparison of the baseline characteristics (**Figure R2** below). There are no significant differences for covariates of interest before TLC-2716 treatment.

Figure R2: Boxplots showing the plasma lipid-related parameters at the start of TLC-2716 treatment in humans.

(2). The 14-day relative (%) change was calculated as $(\text{Day 14} - \text{Day 1}) / \text{Day 1}$ and differences between the placebo and each dose group were evaluated. This approach helps to eliminate the influence of differences in baseline parameters.

We have now improved the figure legend to make it clearer: see page 25, line 757-758 in the revised manuscript and page 9, line 117 in the revised Extended Data Figures and Figure Legends file figure legend.

(2) There were some mild adverse events reported, it would be great to conduct a sensitivity analysis by excluding those adverse events to examine the robustness of results.

Thank you for your suggestion. We excluded four individuals with treatment-related adverse effect (see the supplementary table 1) from our 50 participants in the Multiple-Ascending Dose (MAD) cohorts. The effect of TLC-2716 was then evaluated in the remaining 46 individual-level datasets. As shown in below **Figure R3**, this sensitivity analysis revealed a reduced effect of the highest dose (12 mg) of TLC-2716, while the relative changes of small LDL particle number in 14 days treatment are now suggestively decreased (BH-adjusted P value < 0.1) upon 6 and 12 mg TLC-2716 treatment. The significance of the other comparisons was consistent with the results from the full analysis including all individuals (**Figure 5**, page 42 in the revised manuscript). Taken together, this sensitivity analysis highlighted the robustness of our results.

Figure R3: Dot plots showing the therapeutic effect of 14 days feeding of TLC-2716 at 0.5 mg, 2 mg, 6 mg, or 12 mg on plasma lipid biomarkers. * represents the significance of comparisons between Day 14 and Day 1 within the same dose group calculated by Wilcoxon signed-rank test and adjusted by BH method, while # indicates the significance of relative (%) changes of each parameter upon 14 days of treatment compared to placebo calculated by $[(\text{Day 14} - \text{Day 1}) / \text{Day 1}]$ calculated by Mann-Whitney U tests and adjusted by BH method. */#p<0.05; **/##p<0.01; ***/###p<0.001.

5. Multiple comparison: there were many places that involved potential multiple comparison. Although authors indicated significance by use different notations to indicate different levels of significant, it is important to perform multiple comparison adjustment to avoid inflated type I error. Additionally, it would be great to report p-value or/and adjusted p-value as well. Thank you for pointing this out. We have now applied corrections for multiple comparisons where applicable and indicated it in the figure legends. The p-value or/and adjusted p-value for each figure panel was reported in the source data with suffix as result.

Minor:

1. Typo “sush” in “...sush as Fasn, Scd1, and Acaca...”.

This was corrected in the manuscript (page 5, line 131).

2. Phase 1 clinical trial section, a citation might be needed for “TLC-2716 was first tested in a single-ascending dose (SAD) study in healthy subjects in which it was shown to be safe and well tolerated in single doses up to 20 mg.”

We have added a citation to indicate TLC-2716 is the first LXR inverse agonist tested in humans, please find it in the revised manuscript (page 10, line 303-304).

3. GWAS study signals are identified using widely use p-value threshold of 5e-08 to consider for multiple comparisons. In the method section, authors used 1e-08 as the threshold. It would

be great that author can justify why a more stringent threshold was used here. A more stringent threshold might lead to missing signals. Please clarify the choice of this arbitrary threshold.

We use a more stringent threshold to obtain the most significant associations. Therefore, we choose 1e-08 as threshold. Based on this stringent threshold, we already observed association between lipid-related parameters and *LXRα* or its target genes, indicating the importance of the LXR pathway in lipid metabolism.

We though agree with the reviewer that a more stringent threshold could obscure some signals. Therefore, we have now applied 5e-08 as threshold for the GWAS study based on UKBB, the FinnGen, and the MVP database. This less stringent threshold does not affect the results a lot and only plasma glucose and hyperglyceridemia as additional signals were observed in the UKBB study and the FinnGen study, respectively, but did not change the result in the MVP database. In addition, we also include hypertension and body composition in the UKBB GWAS study. The parameters we used for the UKBB are also listed in the source data. Please find the updated **Figure 1a** in the revised manuscript (page 38) and **Extended Data Figure 1b** in the revised Extended Data Figures and Figure Legends file (page 1).

In all, this study presented compelling evidence that the gut- and liver-restricted LXR inverse agonist TLC-2716 improves lipid metabolism and showed promise as a therapeutic strategy for dyslipidemia and residual ASCVD risk. While the study was well-designed and provided valuable preclinical and early clinical insights, several major issues limited the strength of the conclusions, including the need for additional post-GWAS analyses (fine-mapping, TWAS, Mendelian randomization), clarification and refinement of statistical models in animal studies, baseline and covariate adjustments in the clinical trial, detailed description of phase I clinical trial results, and proper multiple-comparison corrections. Addressing these concerns, along with minor revisions for clarity and consistency, would greatly strengthen the manuscript and its translational impact.

We thank you for your encouraging comments.

Reviewer #2 (Remarks to the Author):

The uniqueness of this study is based on the irony of the efficacy (in humans) of compounds that target LXR for suppression of LXR target genes (LXR inverse agonists), rather than LXR agonists that have been a major focus of therapeutic development for decades. Beginning in 2000, LXR was shown to increase ABCA1 expression and reverse cholesterol transport (RXT) (PMID: 10858438; PMID: 11790770) and in 2002 LXR agonists were first shown to have efficacy in reduction of atherosclerosis (PMID: 12032330) leading to a substantial effort to develop such synthetic agonists to treat heart disease. However, about the same time, synthetic LXR agonists were also shown to increase hepatic lipogenesis leading to fatty liver and elevated plasma levels of TGs via induction of key genes such as SREBP1, FAS, and SCD1 (PMID: 11090131). This side effect limited the potential clinical utility of LXR agonists. This

led to years of attempts to develop LXR agonists that could increase RCT while having no effect on fatty liver and TGs. LXR agonists with favorable preclinical profiles were developed but displayed challenging data in humans indicating that although LXR agonism drove an increase in HDL-C, it also was associated with an increase in TGs and unexpectedly, an increase in LDL-C as well (PMID: 27508871). In 2013, with the knowledge that LXR activation was very effective in induction of lipogenesis, the Burris group focused on developing the first LXR inverse agonists that would selectively recruit corepressors to LXR resulting in suppression of LXR target genes leading to reduction in hepatic lipogenesis. Understanding that these compounds would likely also inhibit RCT they developed tool compounds that displayed liver selectivity, and these compounds were efficacious in animal models of fatty liver (PMID: 27508871), MASH (PMID: 25830098), and ASH (PMID: 31696159; PMID: 33770118) without affecting RCT in the periphery. These first-generation compounds did not have oral bioavailability, but demonstrated the feasibility of a liver targeted LXR inverse agonist. These first-generation compounds also had beneficial effects on insulin sensitivity, plasma TGs, and LDL-C levels (PMID: 25830098; PMID: 35417135).

In this manuscript, Li et al describe the profile and development of LXR inverse agonist for treatment of dyslipidemia and MASH and results of clinical data demonstrating their effectiveness in humans. The manuscript describes the preclinical development of 2 LXR inverse agonists and further clinical data (phase 1) with one selected LXR inverse agonist with a favorable PK/PD and toxicology profile. Preclinical data includes an array of biochemical, cell based, and rodent models. The profile is impressive and the lead compound that proceeded to clinical trials displayed outstanding safety data, selective liver exposure, and outstanding efficacy in lowering plasma TG and LDL-C. Beyond the expected suppression of genes that encode proteins essential for lipogenesis, a very interesting effect of this class of compound is its pleiotropic effect on expression of a range of genes whose products are targets for dyslipidemia themselves including HMGCR, PCSK9, IDOL, ANGPTL3, and APOC3. There is significant promise for such a compound for treatment of dyslipidemia, fatty liver and MASH in humans and the results are quite exciting. Below are several areas that should be addressed:

1. In the introduction, the authors use the term “downregulation” of LXR activity, and this term is not a sufficient method to describe an inverse agonist. Downregulation suggest that the effect of a compound is to reduce the efficacy of activation of a receptor. This is not what is occurring since the inverse agonist is causing an active repression of target genes and not merely downregulating the ability of genes to be activated.

Thank you for your suggestion, we have replaced “downregulation” by “repression” as you suggested. Please find the revised text in the manuscript.

2. No mention of previous development of LXR inverse agonists or liver selective LXR inverse agonists is mentioned in the introduction. As mentioned above, the preclinical proof-of-principle of liver selective LXR inverse agonists has been conducted in models of MASH and dyslipidemia previously. This should be referenced in the introduction.

Thanks for pointing this out and for the nice succinct overview of the development of the LXR compounds given in the pre-amble. We have now specified the development of LXR inverse agonists in the introduction according to the information you provided (see page 4, line 80-87).

3. On line 130, the authors describe the TLC compounds as “first-in-class” LXR inverse agonists. These are not the first LXR inverse agonists discovered and tested in preclinical models, but one is the first in humans. This should be clarified.

Thanks again, we have now removed the “first-in-class” in that sentence and then clarify that TLC-2716 is the first LXR inverse agonist tested in humans in the introduction (see page 4, line 91) and in the opening paragraph in the discussion section (see page 12, line 379-381).

4. I’m unable to find the specifics of how many of the biochemical and cell-based assays were performed. The binding assay and transfection assays have no methods. Were the binding assays radioligand binding displacement assays? If so, what was the radioligand utilized to displace? K_i values would be more appropriate than EC_{50} values. It would also be useful to have a comparator compound such as SR9238 to compare the activity of the TLC compounds to in these assays.

The methods have been updated to include description of the biochemical and cellular potency assays, please find it in the revised manuscript (page 28). The biochemical assays employed TR-FRET-based evaluation of displacement of a N-terminally biotinylated co-activator NCOA3 to recombinant GST-tagged LXR- α or - β ligand-binding domain. These assays were run during the lead optimization phase of the program, and comparison to other LXR inverse agonists was not performed. However, the reported IC_{50} for SR9238 in a similar assay evaluating recruitment of the co-repressor NCOR1 was 33 and 13 nM for LXR- α and - β , respectively³, and this is similar to the potency of TLC-2716 and TLC-6665 reported here. Moreover, the assay differentiated between compounds with varying potencies as has been reported in the compound patent⁴.

5. In the evaluation of the 2716 in the rodent models (beginning with line 142) was there plasma exposure data used to select the doses of 0.3 and 1 mg/kg/day. Given the relatively short half-life, was the compound ever examined with b.i.d. dosing?

Based on the data in **Figure 3g**, we can see that despite low levels in plasma, the levels of TLC-2716 in the liver are sustained over the course of 24h. Therefore, more frequent dosing the TLC-2716 is not expected to provide additional benefit and was therefore not examined.

6. On line 255, the authors describe the liver selective exposure. Have liver partition coefficients been calculated (K_p , liver) for each of these compounds?

Based on the PK data shown in the **Figure 3g**, the liver partition coefficients (K_p) is 143-856 for TLC-2716 and 55-168 for TLC-6665 from 0 to 24 hours after compound exposure in mice. This results further demonstrate the accumulation of TLC-2716 in the liver.

7. In Figure 3i, it appears that there are two labels for the y axis- Fasting Glucose and mRNA Thank you for pointing this out, we have corrected this in the revised Extended Data Figures and Figure Legends file (**Extended Data Figure 5j**, page 5).

8. In extended figure 6a, it appears that there is a very large food effect on plasma drug levels (based on the 6 mg dose). Was this the case? If so, was this expected based on the previous PK studies?

In the clinical study, we observed that fasting can improve absorption of TLC-2716. However, we did not evaluate the impact of food intake on plasma TLC-2716 levels in mice, as such effects are usually species-specific.

Reviewer #3 (Remarks to the Author):

I read with interest this investigation of LXR in lipid metabolism, insulin sensitivity and metabolic liver disease by Li and colleagues. They have presented a comprehensive series of studies that include genome wide association studies from multiple cohorts, metabolic mouse models of both dyslipidemia and liver disease and then studies leading to a phase 1 trial of a LXR inverse agonist. Essentially the findings consistently implicate LXR α in lipid metabolism, favorable effects in mouse models and organoid studies on dyslipidemia, insulin sensitivity and MALSD and a phase 1 trial demonstrating the LXR inverse agonist being well tolerated, having beneficial effects on atherogenic lipid parameters and improving surrogate markers of insulin sensitivity. The selective inverse agonist designed to work within the liver and gut, not systemically in order to avoid potential adverse effects on reverse cholesterol transport. The strength of the manuscript lies in the comprehensive series of studies presented. The authors should consider a number of points.

We thank the reviewer for the positive evaluation of our work.

1. There is a large use of surrogate readouts. A lot of gene regulation results, without accompanying functional data – the reverse cholesterol transport data is a good example, with only ABCA1 presented as if that is a perfect correlate of transport, which it is not – similarly, the surrogates of insulin sensitivity in the phase 1 trial. There are many more. This in my mind softens the strength of the paper.

Thank you for this comment. Unfortunately, in some cases only surrogate readouts are possible. LXR being a nuclear receptor, in preclinical studies, gene expression is typically used as a direct read-out of LXR activity in various tissues^{2,5}. LXR mediated inhibition of target genes in macrophages would indicate systemic effects of TLC-2716, and therefore may be expected to decrease reverse cholesterol transport. The fact that we did not observe a reduction in the expression of LXR target genes in the buffy coat would suggest no direct LXR inhibition in the periphery, and therefore no effect on RCT would be expected. We agree that this is a surrogate readout and a limitation, and we now acknowledge this in the text in page 14, lines 473-484.

2. The field of LXR agonists has proven challenging in drug development, largely due to broad pleiotropic effects, safety issues and modest metabolic efficacy. The authors imply that a potential adverse effect on reverse cholesterol transport is the problem – it is ONE problem. A more balanced view of the previous challenges in this field and how they propose to distinguish the findings from those programs would be important to do.

Thank you for your suggestions. Reviewer #2 also indicated the same point. Based on the preamble of Reviewer #2 and your comments, LXR agonists with favorable preclinical profiles were developed but LXR agonism not only drove an increase in HDL-C, but it also was associated with an increase in TGs and unexpectedly, an increase in LDL-C as well⁶. Therefore, it has been challenging to develop LXR agonists with reduced side effects to treat cardiovascular disease in humans.

With the knowledge that LXR activation was very effective in induction of lipogenesis, the first LXR inverse agonists were developed that would selectively recruit corepressors to LXR resulting in suppression of LXR target genes leading to reduction in hepatic lipogenesis. Knowing that these compounds would likely also inhibit RCT, tool compounds that displayed liver selectivity were developed, and these compounds were efficacious in animal models of fatty liver⁶, MASH⁷, and ASH^{8,9} without affecting RCT in the periphery. These first-generation liver specific LXR inverse agonists did not have oral bioavailability, but demonstrated the feasibility of developing a liver targeted LXR inverse agonist. This first-generation compound and another gut-specific LXR agonist also showed beneficial effects on insulin sensitivity, plasma TGs, and LDL-C levels^{7,10}.

Differently from these LXR inverse agonists, our compound, TLC2716, is a liver- and gut-restricted orally administered LXR inverse agonist, which is well tolerated, and produces significant improvements in plasma TG, remnant cholesterol, LDL particles, and other atherogenic lipids based on preclinical experiments and a phase 1 clinical trial.

Based on your and #Reviewer 2's suggestion, we have now improved our introduction to list the previous challenges in the development of LXR agonists and inverse agonists. Please find the revised text in the introduction section of manuscript (see page 4, line 80-87).

3. The authors should tone down comments as “safety” in the phase 1 trial. They would be better to talk to tolerability and the presence or lack of adverse events.

Thank you for pointing this out, we have now changed “safety” to “tolerability”. Please find the modified text in the revised manuscript.

4. The individuals selected for inclusion in the phase 1 trial had modest dyslipidemia at best. Given the range of triglyceride levels at baseline, none of these individuals would be candidates for this drug.

Since subjects eligible for this trial were healthy volunteers, their lipid values at baseline were at most mildly elevated. We took this approach because the primary objective of the Phase 1 trial was to assess the tolerability and PK of TLC-2716 in healthy individuals. Nevertheless, an attempt was made to enroll subjects in the MAD cohorts with TG \geq 150 mg/dL and/or LDL-C \geq 130 mg/dL (see Methods: Inclusion and Exclusion Criteria, page XX in the revised manuscript) to enable preliminary assessment of the lipid-lowering benefits of TLC-2716. In the ongoing Phase 2 clinical trial, we enrolled a larger population of patients with dyslipidemia (TG \geq 350 mg/dL) to further evaluate the therapeutic potential of the compound. This point is now also included in the section describing limitations of this study (page 14, lines 473-484).

5. The findings are modest. The triglyceride lowering is far below what is being shown with inhibitors of ANGPTL3 and apoC3. The apoB reduction is far below what would be needed to progress this compound in development. The Mendelian randomization data is clear in demonstrating the relationship between apoB lowering and CV benefit in this space. I'm not convinced that there is enough here to be too optimistic that this agent will be a useful therapeutic in the prevention clinic. Much more work is required.

We agree with the reviewer that extensive additional work is required to determine the clinical potential of TLC-2716 for lipid lowering in patients with dyslipidemia. Indeed, this Phase 1 study in **healthy volunteers** is the first step in clinical development. As described, a Phase 2 trial in **patients with severe hypertriglyceridemia and metabolic dysfunction-associated steatotic liver disease (MASLD)** is ongoing. Notably, our preclinical (**Figure 2e**) and clinical data (**Extended Data Figure 8c**) demonstrate that TLC-2716 has a more profound effect in settings with elevated baseline TG, likely due to upregulation of the LXR pathway.

We also agree that inhibitors of ANGPTL3 and apoC3 are very promising. While these compounds act primarily on the liver to reduce TG-rich lipoproteins, the mechanism of action of our LXR inverse agonist differs in several ways:

- (1). TLC-2716 acts beyond the modulation of one or two genes. In addition to lowering ANGPTL3 and apoC3, it has pleiotropic hepatic and intestinal effects to suppress DNL in both tissues, hepatic VLDL secretion and cholesterol synthesis, and intestinal lipid absorption, as well as increase clearance of VLDL and LDL-C. These pleiotropic mechanisms contribute to the observed reductions of lipid parameters in our preclinical and clinical studies. This may limit loss of efficacy via compensatory pathways and potentially enhance safety as complete inhibition of a particular molecular target may pose safety risks and not be necessary for maximal efficacy.
- (2). Unlike ANGPTL3 and apoC3 inhibitors currently in development, which require parenteral administration, TLC-2716 has the advantage of oral administration. Many patients find this preferable, even if it comes with somewhat reduced efficacy (see page 14, lines 470-472 in the revised manuscript). Moreover, an added advantage of an oral compound is that it could be combined in fixed-dose combinations with other approved oral agents (e.g., statins, ezetimibe, and fibrates) which can ease pill burden for patients.

References:

1. Cui, R. *et al.* Improving fine-mapping by modeling infinitesimal effects. *Nat. Genet.* **56**, 162–169 (2024).
2. Lockhart, S. M. *et al.* Damaging mutations in liver X receptor- α are hepatotoxic and implicate cholesterol sensing in liver health. *Nat. Metab.* **6**, 1922–1938 (2024).
3. Griffett, K., Solt, L. A., El-Gendy, B. E.-D. M., Kamenecka, T. M. & Burris, T. P. A Liver-Selective LXR Inverse Agonist That Suppresses Hepatic Steatosis. *ACS Chem. Biol.* **8**, 559–567 (2013).
4. Gege, C. *et al.* LXR modulators with bicyclic core moiety. (2024).
5. Clark, A. T. *et al.* A mutation in LXR α uncovers a role for cholesterol sensing in limiting metabolic dysfunction-associated steatohepatitis. *Nat. Commun.* **16**, 1102 (2025).
6. Kirchgessner, T. G. *et al.* Beneficial and Adverse Effects of an LXR Agonist on Human Lipid and Lipoprotein Metabolism and Circulating Neutrophils. *Cell Metab.* **24**, 223–233 (2016).
7. Griffett, K. *et al.* The LXR inverse agonist SR9238 suppresses fibrosis in a model of non-alcoholic steatohepatitis. *Mol. Metab.* **4**, 353–357 (2015).
8. Sengupta, M., Griffett, K., Flaveny, C. A. & Burris, T. P. Inhibition of Hepatotoxicity by a LXR Inverse Agonist in a Model of Alcoholic Liver Disease. *ACS Pharmacol. Transl. Sci.* **1**, 50–60 (2018).
9. Sengupta, M. *et al.* A two-hit model of alcoholic liver disease that exhibits rapid, severe fibrosis. *PloS One* **16**, e0249316 (2021).
10. Griffett, K. *et al.* Antihyperlipidemic Activity of Gut-Restricted LXR Inverse Agonists. *ACS Chem. Biol.* **17**, 1143–1154 (2022).

Reviewer #4 (Remarks to the Author):

The authors are thanked for their responses. I remain concerned that the majority of findings are indirect at best and the phase 1 findings are underwhelming.

Thanks for your comments. Our extensive preclinical evidence has provided support for our Phase 1 study where we show that the compound was well-tolerated and induced dose-dependent lipid reductions after 14 days of treatment, consistent with its mechanism of action.

In addition, we used gene expression analysis in buffy coats to evaluate the effect of TLC-2716 on reverse cholesterol transport in mice and humans. We agree that this is a surrogate readout and a limitation of our work, that we now acknowledge this in the discussion.

As we mentioned in the previous rebuttal letter, this Phase 1 study in **healthy volunteers** is the first step in clinical development, while a Phase 2 trial in patients with severe hypertriglyceridemia and metabolic dysfunction-associated steatotic liver disease (MASLD) is ongoing. Notably, our preclinical (**Figure 2e**) and clinical data (**Extended Data Figure 8c-e**) demonstrate that TLC-2716 has more profound effects in settings with elevated baseline levels of TGs, remnant cholesterol, and LDL-C. We therefore also agree with this reviewer that the results obtained from Phase 2 trial will better reflect the effect of TLC-2716 on lipid reduction in patients.

Overall, our preclinical studies with sufficient evidence have demonstrated the effect of TLC-2716 on improving plasma lipid metabolism by repressing hepatic LXR activity, while the Phase 1 clinical trial has shown that TLC-2716 is safe and well-tolerated, with lipid reductions observed within 14 days of treatment.